# COVID-19 Pandemic Response Robot

**Min-Fan Ricky Lee** [1,2,*] and **Yi-Ching Christine Chen** [1]

1   Graduate Institute of Automation and Control, National Taiwan University of Science and Technology, Taipei 106335, Taiwan; m10712026@mail.ntust.edu.tw
2   Center for Cyber-Physical System Innovation, National Taiwan University of Science and Technology, Taipei 106335, Taiwan
\*   Correspondence: rickylee@mail.ntust.edu.tw

**Abstract:** Due to an arising COVID-19 positive confirmed case in Taiwan, the screening of body temperature, mask wearing and quarantined violation is enhanced. A mobile robot that conducts this task is demanded to reduce the human labor. However, conventional robots suffer from several limitations, perceptual aliasing (e.g., different places/objects can appear identical), occlusion (e.g., place/object appearance changes between visits), different viewpoints, the scale of objects, low mobility, less functionality, and some environmental limitations. As for the thermal imager, it displays the current heat spectrum colors, and needs manual monitoring. This paper proposes applying Simultaneous Localization and Mapping in an unknown environment and using deep learning for detection of temperature, mask wearing, and human face on the Raspberry Pi to overcome these problems. It also uses the *A\** algorithm to do path planning and obstacle avoidance via 3D Light Detection and Ranging to make the robot move more smoothly. Evaluating and implementing different Simultaneous Localization and Mapping algorithms and deep learning models, then selecting the most suitable method. Root Mean Square Error of three Simultaneous Localization and Mapping algorithms are compared. The predictions of deep learning models are evaluated via the metrics (model speed, accuracy, complexity, precision, recall, precision–recall curve, F1 score). In conclusion, Google Cartographer for building a map, Convolutional Neural Network for mask wearing detection, and only looking once for human face detection achieve the best result among all algorithms.

**Keywords:** artificial intelligence; deep learning; disaster response; mobile robots; navigation

## 1. Introduction

At this stage, the epidemic is severe around the world and the infection rate is very high. Nowadays, people's awareness of their own health is gradually improving; it is a fact that researchers have shown an increased interest in epidemic prevention. With new technology continuously advancing, mobile robots are widely used in many different fields, such as logistics services, industrial transportation system, etc. Recent developments in the field of artificial intelligence have led to a renewed interest in medical combined with artificial intelligence. However, conventional robots suffer from several limitations, perceptual aliasing, occlusion, different viewpoints, the scale of objects, low mobility, less functionality, and some environmental limitations. In previous papers, most methods of taking people's temperature via a thermal imager could do preliminary screening effectively. Nevertheless, the previous experiment results were not enough for us. According to previous papers, the mobility of the thermal imager is one of the significant limitations in those methods. Most of the thermal imagers at many quarantine stations are installed in fixed areas. The location of the thermal imagers is at the entrance and the view point is fixed. Therefore, those methods cannot adapt to various environments. Some existing algorithms have proven to accurately measure people's temperature through the thermal imager, but they still have less function, are easily affected by the temperature of other objects, and cannot move. Furthermore, it can only display the current heat spectrum

colors, and needs a person to monitor. There is a significant difference between manual monitoring and machine detection. The error rate of manual monitoring is higher than machine detection. In terms of the detection of mask wearing and human face, they may be affected by light, the scale of objects, people's poses, obstructions, etc. Hence, it is necessary to use an alternative method.

Why are the above-mentioned problems and the epidemic prevention issues important? According to the data of Taiwan Centers for Disease Control, the highest attendance rate of weekly emergency-type influenza is 22.22% in the past two years. It is still 6.46% so far. In addition to flu, there are other diseases with higher infection rates and some diseases may be asymptomatic infection. Through the literature review, the problems in the previous paragraph can be proved to be important. The Gmapping package was used on both PC and Raspberry Pi to maintain high performance [1]. To solve the problem of lack of environment features, an improved Hector SLAM (Simultaneous Localization and Mapping) algorithm was proposed. This algorithm provides the precise input and projects the scan information onto grid map during scan matching process [2]. An analysis of Gmapping, Hector SLAM, and Google Cartographer was presented and realized in the indoor environment [3]. Therefore, SLAM plays an important role in this system. The use of autonomous mobile robots turns the fixed-point detection into 360-degree surround detection at any time. The advantage of this system is that it can adapt to changing environments by modifying the robot system's configuration. Moreover, combining data from different kinds of sensors in this system can provide users with more accurate information to achieve better results. A physiological monitoring system for assistive robots using a thermal camera is proposed in [4]. The human face detection is affected by various poses, occlusions, lighting, and so on. A novel deep cascade convolutional network was proposed to solve these problems, using the Fast Region-based Convolutional Neural Network (Fast R-CNN) at the end of the structure. The result shows 91.87% recall [5]. In addition, the impact of using various color spaces on the performance of the Faster Region-based Convolutional Neural Network (Faster R-CNN) night human detection was discussed [6]. As for the thermal image, some previous papers apply deep learning to process images. For example, using Faster R-CNN to handle thermography images [7], and thermal image recognition based on You Only Look Once (YOLO) can detect temperature accurately [8] to identify power facilities and to avoid accidents. These approaches are more efficient and accurate than human beings. In order to reduce the fatal injury rate for the construction industry, a perception module was proposed, it used end-to-end deep learning and visual SLAM for an effective and efficient object recognition and navigation via a mobile robot [9].

This paper integrated temperature detection, mask wearing detection, and human face detection to save manpower to see the detection results on the monitor. At the same time, it can also immediately identify the person suspected of hyperthermia and whether the person is wearing a mask or not. Moreover, integrating these identification functions on a mobile robot greatly improves the mobility during detection.

In recent years, people's health awareness has risen, and epidemic prevention equipment has continued to improve. In addition to meeting people's needs, it can also effectively reduce the risk of infection. However, for temperature detection, there are some requirements of camera types to ensure that the measurement results are accurate. General cameras do not have the function of measuring temperature. In terms of mobility, the single thermal imager cannot move to anywhere; it can only be installed in a fixed position. While some articles use thermal cameras in areas such as security robots and guide robots [10], this provide only humans' temperature, which is not a sufficient biometric measure in most of the cases. This paper addressed previous unanswered questions and proposed an alternative method. In order to realize the intelligence of the mobile robot, autonomous localization, mapping, and obstacle avoidance cannot be lacking. Light Detection and Ranging (LiDAR) with SLAM was used for autonomous navigation by a mobile robot. When navigating the mobile robot, it needs to obtain the map of the environment in advance for path planning; thus, SLAM has been widely studied to solve this problem. The method of SLAM starts

from an unknown location in an unknown environment; the mobile robot locates its own position and posture by repeating the observed map features during movement, and then incrementally constructs a map based on its own position to achieve the purpose of simultaneous positioning and map construction. There are some different approaches for object detection and face detection now and they have been implemented and verified. This paper extends previous papers, various deep learning frameworks, and evaluation indicators, including Convolutional Neural Network (CNN), Region-based Convolutional Neural Network (R-CNN), mask Region-based Convolutional Neural Network (Mask R-CNN), and YOLO, which have been applied. Hence, they can be used for the best model selection.

This paper proposes an approach to verify three detections (temperature, mask, face) in real time by using deep learning models on the mobile robot. Nowadays, deep learning in object detection and face detection has grown significantly and it has been applied to access control system. Compared with conventional solutions based on manual feature extraction, using deep learning solutions is better in many monitoring systems. For the conventional approaches, the result of feature extraction depends on the engineer's skills. In contrast, for deep learning approaches, the design of manual features is not needed and people can feed the original image data directly to a deep learning structure to achieve detection. The aim of this paper is to evaluate and validate the above verification experiment, which is divided into two stages. First, it is necessary for the mobile robot to carry out self-positioning and navigation in an unknown indoor environment. Second, detecting accurately in real time and obstacle avoidance are indispensable while the mobile robot is moving. This paper applied three SLAM algorithms, which are Gmapping, Hector SLAM, and Google Cartographer. After evaluation and testing, the best algorithm was selected from these three algorithms. When selecting the SLAM algorithm, Root Mean Square Error (RMSE) was used as the consideration and standard. Among these three algorithms, Google Cartographer was chosen because it shows the minimum RMSE and the processing time is acceptable. Next, for the path planning part, *A** algorithm was used. Finally, in order to achieve navigation, an algorithm called Adaptive Monte Carlo Localization (AMCL) was used. To conclude, the mobile robot in this paper operates on a Robotic Operating System (ROS). The mobile robot performs SLAM with LiDAR to obtain the matching data of LiDAR scanning and Google Cartographer SLAM. In an indoor environment, mapping, navigation, path planning and obstacle avoidance indoor via a 3D visualization tool for ROS called RVIZ are implemented. The experimental results show that three SLAM algorithms are feasible. For the detection part, image frames are put into the deep learning models. The accuracy and recall of the results are verified to select the most suitable model.

## 2. Materials and Methods

This part introduces the methods to solve the problems mentioned before. Mainly SLAM algorithms are used to build a map in an indoor environment, the *A** algorithm for path planning, and the AMCL to let the mobile robot self-position and navigate. Deep learning is applied for detection of temperature, mask wearing, and human faces. The action of the mobile robot in an indoor public place is presented in Figure 1.

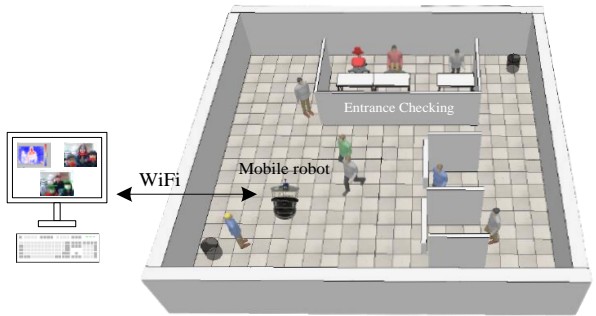

**Figure 1.** Scenario diagram.

This paper proposes an intelligent behavior control system for multiple mobile robots, as shown in Figure 2. The autonomous robotic behavior includes obstacle avoidance, goal seeking, trajectory tracking, and formation. The frontier command is located on the site workstations for collecting raw sensory data and sending task commands to robots. The remote Artificial Intelligence (AI) strategy is in the cloud for highest strategical decision making.

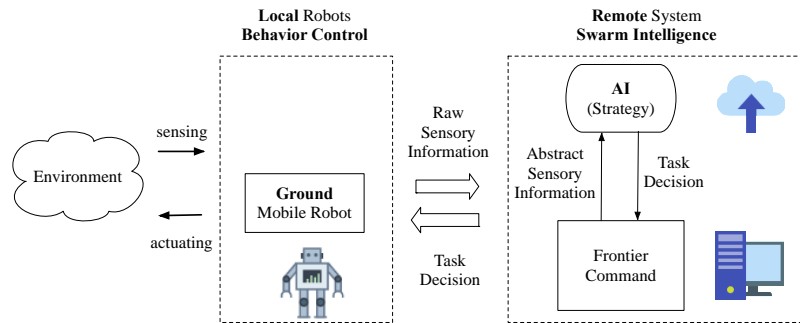

**Figure 2.** System: Remote swarm AI and local robotic behavior control.

The proposed Artificial Intelligence of Things (AIoT) architecture is illustrated in Figure 3. The workflow is as follows:

1. Data Collection: sensory data from the environment are transmitted to the cloud through the site workstation.
2. Training: The AI model is built using the data collected by mobile robots from the environment.
3. Validation: The AI model is evaluated for hyper-parameter tuning and model selection.
4. Testing: The trained AI model is further deployed to the local workstation for further testing the performance. The tested model is finally deployed to the embedded system in mobile robots and the learning process is continued.

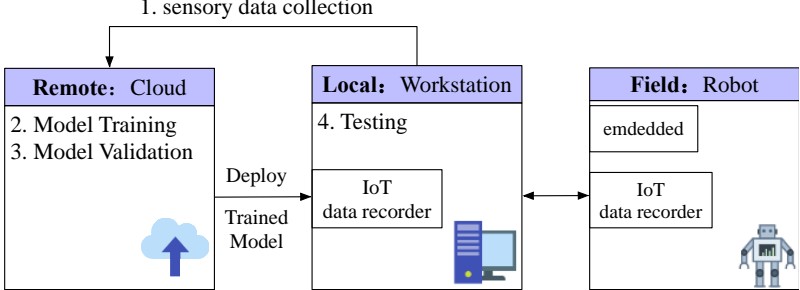

**Figure 3.** AIoT architecture.

The hierarchical control architectures of autonomous mobile robots proposed in this paper are shown in Figure 4. The high-level behavior control system is located at the remote end and the low-level pose control system is located at the mobile robot in the field. The communication includes wireless (distributed) and wire (centralized).

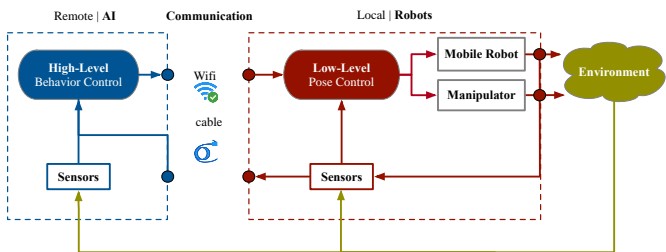

**Figure 4.** Hierarchical control: High level and low level.

The overall system architecture is shown in Figure 5. The map is obtained through SLAM, and the robot is applied to the actual indoor environment by path planning and navigation. At the same time, the camera is used to do real-time detection.

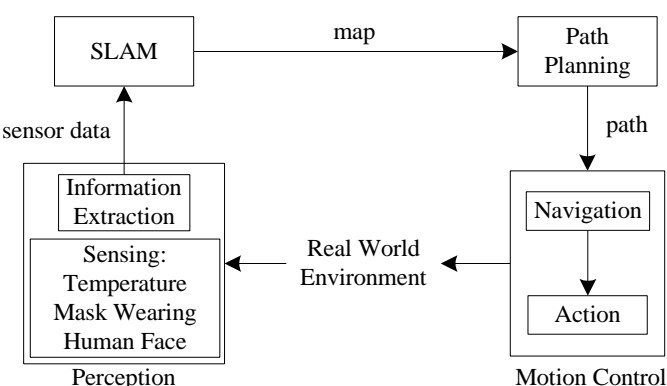

**Figure 5.** System architecture.

### 2.1. SLAM

In this section, three common SLAM algorithms and navigation methods are introduced. This paper selected three of the most common SLAM algorithms, which are Gmapping, Hector SLAM, and Google Cartographer [11]. Due to the need for a set of evaluation criteria to ensure accuracy, a navigation system based on SLAM was designed in this paper. The process was as follows. First, the mobile robot received data from 3D LiDAR. According to these data, the mobile robot could calculate its own position and build a map of the environment through the SLAM algorithm. At the beginning of navigation, this paper used AMCL to locate the initial pose of the mobile robot. Next, for the path planning part, the *A** algorithm was used to generate the global path. Finally, in the part of the obstacle avoidance LiDAR was used to implement it. Several laser-based 2D SLAM techniques were evaluated according to reviewed literature. This paper selects three of the most popular and commonly used SLAM algorithms to evaluate each performance. In order to pursue the accuracy of the evaluation criteria, this paper designed a consistent SLAM-based navigation system. The off-line and real-time map building is illustrated in Figure 6. The on-line and real-time SLAM and object detection is illustrated in Figure 7.

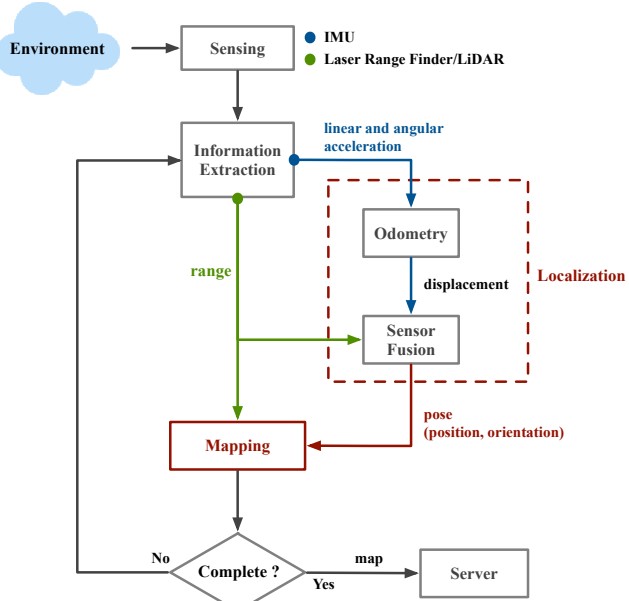

**Figure 6.** Off-line map-building workflow.

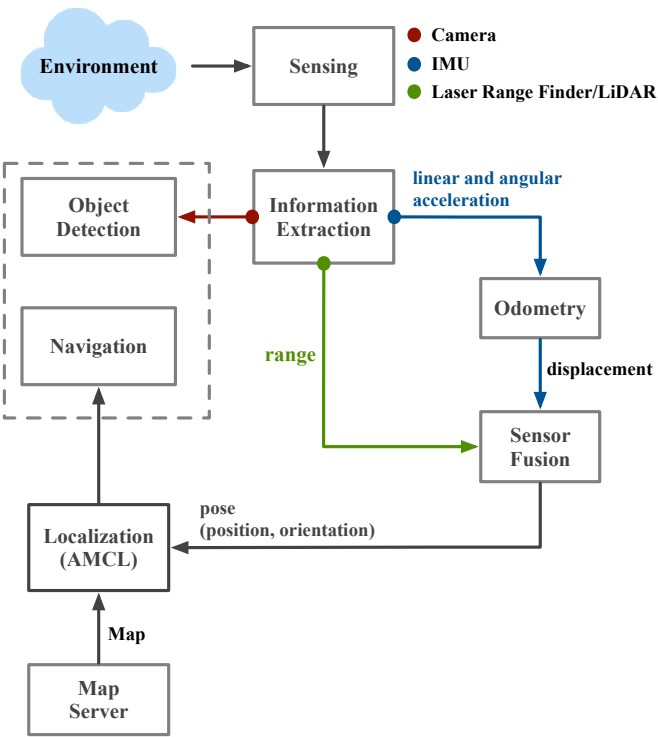

**Figure 7.** Off-line map-building workflow.

### 2.1.1. Gmapping

Gmapping is a common open-source SLAM algorithm based on the filtering SLAM framework. There are some serious depletion and computational complexity problems for the conventional particle filter. It is based on the Rao-Blackwellized Particle Filter (RBPF) [12,13] particle filter algorithm, which separates the positioning and mapping process, and performs positioning before mapping. Gmapping also uses an adaptive resampling technique. Thus, it solves the previous problem of the particle filter.

The main idea of the RBPF for SLAM is to do the following factorization and estimate a posteriori $p$, as in (1), whereas $p\left(\cdot\mid\cdot\right)$ is conditional probability.

$$p(x_{1:t}|z_{1:t}, u_{0:t}) = p(m|x_{1:t}, z_{1:t})p(x_{1:t}|z_{1:t}, u_{0:t}), \tag{1}$$

where $p$ is a posterior, $x_{1:t}$ is potential trajectories, $z_{1:t}$ is observation of posterior, $u_{0:t}$ is the odometry measurements of $x_{1:t}$, and $m$ is the total number of particles.

The Sampling Importance Resampling (SIR) filter is one of the most common particle filtering algorithms. The sensor observations and the odometry readings are available when using the Rao-Blackwellized SIR filter for incrementally mapping processes. The process can be integrated into the following four steps:

1.  Sampling: The next particle $x_t^{(i)}$ originates from current particles $x_{t-1}^{(i)}$ by using a proposal distribution $\pi\left(x_t \mid z_{1:t}, u_{0:t}\right)$.
2.  Correction: According to the sequence of calibration observations, the corresponding inherent weight is calculated for each particle. Each particle will pair a weight.

$$w^{(i)} = \frac{p\left(x_t^{(i)} \middle| z_{1:t}, u_{0:t}\right)}{\pi\left(x_t^{(i)} \middle| z_{1:t}, u_{0:t}\right)}, \tag{2}$$

3.  Resampling: Particles are drawn with replacement proportional to their importance weight. The particle with a low weight $w$ will be removed.
4.  Map estimation: The corresponding map estimate $m_t^{(i)}$ is calculated from the trajectory and observations for every sampled particle.

Most current applications for particle filters depend on a recursive structure. There is an accurate proposal distribution in this paper and that performs resampling adaptively. Two methods used to develop the Rao-Blackwellized mapping algorithm were used in the adaptive resampling technique. These two techniques improved proposal function, decreased robot pose uncertainty, and used selective resampling to solve the particle depletion problem. The scan-match method estimates the robot position using the gradient descent method, the current generated map, the current LiDAR point, and the robot position as the initial estimates. The scan score is computed as follows:

$$\text{score}(scan, map) = \sum_{p \in scan} e^{-\frac{1}{\sigma} \cdot d(p, map)^2}, \tag{3}$$

where $p$ is a scan point, $d(p, map)$ is the shortest distance between the obstacle stored in the map and $p$, and $\sigma$ is the predefined parameter. Even in large enough environments (250 m $\times$ 250 m), Gmapping only requires between 30 and 80 particles to build accurate maps. Therefore, this method can obtain more precise robot positioning through the sensor data and the movement model.

### 2.1.2. Hector SLAM

Hector SLAM is one of the scan matching algorithms. It combines scanning laser rangefinder matching and the 2D navigation approach with an inertial sensing system by Extended Kalman Filter (EKF). While there is no explicit loop closure detection in this system, the accuracy is high in many practical situations [14]. The EKF can overcome the linearity assumption of the Kalman Filter (KF). The KF model assumes the true state at time $k$ evolved from the state at $(k - 1)$ and the probability of the model's next state must be a linear function $x_k$, as in (4).

$$x_k = F_k x_{k-1} + B_k u_k + w_k, \tag{4}$$

where $F_k$ is the state transition model which is applied to the previous state $x_{k-1}$, $B_k$ is the control-input model which is applied to the control vector $u_k$, and $w_k$ is the random variable representing the prediction noise.

At time $k$ an observation (or measurement) $z_k$ of the true state $x_k$ is made according to (5).

$$z_k = H_k x_k + v_k, \tag{5}$$

where $H_k$ is the observation model which maps the true state space into the observed space and $v_k$ is the observed noise. When the mobile robot is in state $x_k$, the observation will be received ideally.

The EKF is different from the KF. The state transition and observation models do not need to be linear functions of the state, but they may instead be differentiable functions in the EKF; $x_k$ and $z_k$ are shown in (6) and (7).

$$x_k = f(x_{k-1}, v_k) + w_k, \tag{6}$$

$$z_k = h(x_k) + v_k, \tag{7}$$

where $w_k$ and $v_k$ are the process and observation noises. The predicted state and the predicted measurement are the functions $f$ and $h$, respectively.

For scan matching, the Gauss–Newton method was used to solve the problem, which has higher requirements concerning the sensor [15]. The Gauss–Newton iteration method is an iterative method for finding regression parameters for minimum squares in a nonlinear regression model, and its aim is to minimize the error of the occupancy $M(S_i(\zeta))$ to get the end pose. The target function $\zeta^*$ is shown in (8). In this paper, the Gauss–Newton optimization method aligns the laser beam endpoints and the obtained map. A continuous map coordinate $P_m$ is defined using the four closest coordinates $P_{00}, P_{01}, P_{10}, P_{11}$ to approximate

the occupancy value as $M(P_m)$ and the gradient $\nabla M(P_m)$, as in (9). Linear interpolation along the $x$-axis and $y$-axis yields $M(P_m)$, as in (10).

$$\xi^* = \underset{\xi}{\operatorname{argmin}} \sum_{i=1}^{n} \left[ (1 - M(S_i(\xi))) \right]^2, \tag{8}$$

$$\nabla M(P_m) = \left( \frac{\partial M}{\partial x}(P_m), \frac{\partial M}{\partial y}(P_m) \right), \tag{9}$$

$$M(P_m) \approx \frac{y - y_0}{y_1 - y_0} \left( \frac{x - x_0}{x_1 - x_0} M(P_{11}) + \frac{x_1 - x}{x_1 - x_0} M(P_{01}) \right) \\ + \frac{y_1 - y}{y_1 - y_0} \left( \frac{x - x_0}{x_1 - x_0} M(P_{10}) + \frac{x_1 - x}{x_1 - x_0} M(P_{00}) \right) \tag{10}$$

where $n$ is the number of scan readings in each sweep. The $M$ represents map and $\xi = (p_x, p_y, \psi)^{\mathrm{T}}$ is a pose on the map at point $P_{xy}$. The transform of the end-point scan in the robot frame to the world frame is $S_i(\xi)$, as in (11).

$$S_i(\xi) = \begin{bmatrix} \cos(\psi) & -\sin(\psi) \\ \sin(\psi) & \cos(\psi) \end{bmatrix} \begin{bmatrix} S_{i,x} \\ S_{i,y} \end{bmatrix} + \begin{bmatrix} p_x \\ p_y \end{bmatrix}, \tag{11}$$

where $(S_{i,x}, S_{i,y})$ is the laser scan point.

In conclusion, Hector SLAM requires the mobile robot to be controlled at a lower speed because of higher requirements on the sensor. Therefore, the mapping effect will be more ideal. This is also a sequela of no loop closure. When the odometer data is relatively accurate, it cannot be used effectively.

2.1.3. Google Cartographer

Cartographer is part of an open-source SLAM library of ROS, and is developed by Google. The basic architecture of Cartographer includes front-end and back-end. The front-end uses sensor data to create a map, including nodes and edges. The back-end optimizes the map and calculates the best position of the node. It is also a system which provides real-time SLAM in 2D and 3D.

For 2D and 3D internal SLAM, the main sensor relies on LiDAR and inertial navigation [16]. This algorithm uses grid mapping, so it is different from Gmapping and Hector SLAM.

For the scan matching, it is necessary to optimize the scanning pose. The following equation is the scan matching process as it goes through a minimization:

$$\underset{\xi}{\operatorname{argmin}} \sum_{k=1}^{K} \left[ \left( 1 - M_{\mathrm{smooth}}(T_\xi h_k) \right) \right]^2, \tag{12}$$

where $T_\xi$ is the conversion coefficient that shifts a point $h_k$ by $\xi$, $\xi$ is the offset vector $(\xi_x, \xi_y, \xi_\theta)^{\mathrm{T}}$, $M_{\mathrm{smooth}}(x)$ is the value of a cell $x$ smoothed by values in its neighborhood, and $h_k$ is a cell involving a point from the laser scan.

Branch-and-bound scan (BBS) matching is the other method for closed-loop optimization, as in (13).

$$\underset{\xi \in w}{\operatorname{argmax}} \sum_{k=1}^{K} M_{\mathrm{nearest}}(T_\xi h_k), \tag{13}$$

where $w$ is a search window size, $K$ is the last scan point, and $M_{\mathrm{nearest}}$ is the extension of the nearest grid point in $M_{\mathrm{smooth}}$.

The above scan matching method is based on Ceres scan matching. When obtaining each frame data of laser scan, scan matching is used to insert a submap at the best estimated position, and the scan matching is only related to the current submap. It increases the probabilities on a large scale to find a scan pose accurately in the submap.

To conclude, Gmapping is the most widely used algorithm. Gmapping and Cartographer are more accurate than other SLAM algorithms. This paper collects data and experiments in an indoor environment. When selecting the SLAM algorithms, RMSE is

used as the consideration and standard, as in (14). After comparing three algorithms, the most powerful algorithm is Cartographer. Compared to Hector SLAM, Gmapping has lower requirements for LiDAR frequency and high robustness. The result shows that Gmapping and Cartographer have comparable RMSE values on most input sequences. Finally, this paper decided to take advantage of Google Cartographer concerning the mobile robot and construct the map in the indoor environment. The result shows that deformation of the map generated by Cartographer is slightly less than that of the map generated by Gmapping.

$$RMSE = \sqrt{\frac{1}{m}\sum_{i=1}^{m}(y_i - \hat{y}_i)^2},$$ (14)

where $m$ represents the number of different predictions, $y_i$ is the labeled value and $\hat{y}_i$ is the predicted value.

### 2.2. Path Planning and Navigation

Among the above three SLAM algorithms, this paper determines that Google Cartographer with LiDAR is to be used to create a map which is in the indoor environment. After the mobile robot laser scans and builds the map, the indoor map is generated. When obtaining the indoor map, the *A\** algorithm is applied to do global path planning on the map. For local path planning, this paper uses the Dynamic Window Approach (DWA) algorithm to avoid obstacles in real time. According to the result of path planning and obstacle avoidance, the most suitable path is built from the initial point to the target location. The next step is to estimate the position and orientation of the mobile robot, so this paper makes use of AMCL to implement navigation [17]. Navigation workflow architecture is shown in Figure 8. In the process of navigation, the visualization tool for ROS called RVIZ is used at the same time. RVIZ is a 3D visualization tool for ROS applications, and it is also a kind of human–machine interface. It provides viewing of the robot model, capturing sensor information from the sensor on the robot and replaying the captured data. It can display data from cameras, LiDAR, and 2D and 3D devices, including pictures and point clouds.

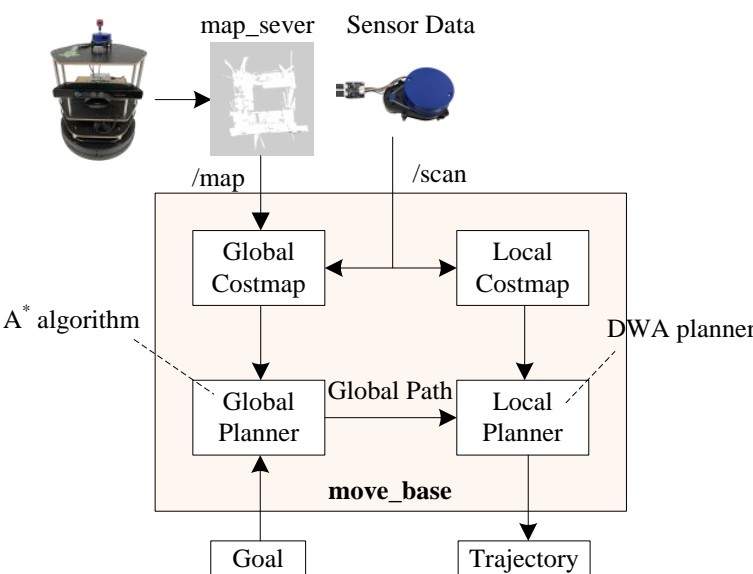

**Figure 8.** Navigation Workflow Architecture.

The algorithm of global path planning is *A\** in this paper; it combines advantages of Dijkstra's algorithm and the Breadth-First Search (BFS) algorithm. Dijkstra's algorithm works by preserving the shortest path of each vertex found so far. The main idea of the Dijkstra algorithm is that the initial point is the center and extends outwards until it reaches the target point. The steps are as follows:

1. First, all nodes are marked as unvisited, and a set is created called the unvisited set.
2. A tentative distance value of each node is set. The value of the initial point is zero, and to infinity for all other nodes.
3. It is important to consider all its unvisited neighbors for the current node and measure their tentative distance within the current node. If the newly allocated distance value is compared with the currently assigned value, then a smaller value is assigned.
4. The current node is marked as visited, after considering all the unvisited neighbors of the current node, and is deleted from the unvisited set. A visited node will not be reviewed again anymore.
5. If the destination node has been marked as visited, or if infinity is the smallest tentative distance between the nodes in the unvisited range, stop.

The BFS algorithm is a strategy for searching in a graph, and it is similar to Dijkstra's algorithm. However, the BFS algorithm can evaluate the cost of any node to the target point; it selects the closest node to the target point.

Due to the completeness, optimality, and optimal efficiency, the *A\** algorithm is applied to many fields. The *A\** algorithm is a graph traversal and path search algorithm, and it does not need to search many nodes. It can also find the shortest path efficiently in the static grid-based map. Because it is difficult for a mobile robot to move in a complex environment, it needs to generate the most suitable route through path planning [18]. After consideration and evaluation, it is feasible to use the *A\** algorithm for path planning. In this paper, the *A\** algorithm is used with 3D LiDAR to do the global path planning and obstacle avoidance, respectively. The *A\** algorithm selects the path that minimizes as $f(n)$, as in (15).

$$f(n) = g(n) + h(n), \tag{15}$$

where $n$ is the next node on the path, $g(n)$ is the cost of the path from the start node to the next node $n$, and $h(n)$ is a heuristic function that estimates the cost of the cheapest path from $n$ to the goal.

The DWA algorithm is used for local path planning in this paper. Due to the speed and acceleration constraints, when a mobile robot is used for obstacle avoidance, its avoidance capability is restricted. Therefore, in the design of dynamic windows, the DWA algorithm takes into consideration the kinematic constraints of finite velocity and acceleration. In the process, the local obstacle avoidance problem of the robot is transformed into a constrained optimization problem in the velocity vector space. This algorithm simulates the moving path of the mobile robot at a certain speed $(v, w)$. After obtaining multiple trajectories, these trajectories are evaluated and the speed corresponding to the best trajectory is selected to let the mobile robot start moving. The evaluation function $E$ is shown in (16).

$$\begin{aligned} E(v, w) =& \alpha \cdot \mathrm{heading}(v, w) \\ &+ \beta \cdot \mathrm{dist}(v, w) + \gamma \cdot \mathrm{velocity}(v, w) \end{aligned} \tag{16}$$

where $\alpha$, $\beta$, and $\gamma$ are weight, heading is a measurement of the deviation of the robot to the target location, dist is the distance to the closest obstacle on the trajectory, and velocity is the forward velocity of the robot.

This paper applies the AMCL algorithm to the indoor mobile robot. AMCL is a probabilistic localization system for a robot moving in 2D [19]. It is simply an upgraded version of the system of Monte Carlo Localization. The current position that is global localization or local localization at first is decided. Next, according to the different status of the position, different particles are assigned to localization and tracking. The MCL algorithm is shown in Algorithm 1 [20]. Given a map of the environment, the goal is for the robot to determine its position within the environment. Every time the algorithm takes as input the previous belief $X_{t-1}$, an actuation command $u_t$, and data received from sensors $z_t$, the algorithm outputs the new belief $X_t$.

| **Algorithm 1** MCL |
|---|
| 1: **function** MCL $(x_{t-1}, u_t, z_t)$ |
| 2: $\qquad \overline{X}_t = X_t = \phi$ |
| 3: $\qquad$ **for** $n = 1$ to $N$: |
| 4: $\qquad\qquad x_t[n] = $ motion_update $(u_t, x_{t-1}[n])$ |
| 5: $\qquad\qquad w_t[n] = $ sensor_update $(z_t, x_t[n])$ |
| 6: $\qquad\qquad \overline{X}_t = \overline{X}_t + < x_t[n], w_t[n]>$ |
| 7: $\qquad$ **end for** |
| 8: $\qquad$ **for** $n = 1$ to $N$: |
| 9: $\qquad\qquad$ draw $x_t[n]$ from $\overline{X}_t$ with probability $\propto w_t[n]$ |
| 10: $\qquad\qquad X_t = X_t + x_t[n]$ |
| 11: $\qquad$ **end for** |
| 12: $\qquad$ **return** $X_t$ |
| 13: **end function** |

AMCL uses the adaptive Kullback–Leibler divergence method to update particles; then it can localize the mobile robot [21]. These particles are provided with their own coordinates and orientation, and have a given weight. Particles with larger weights are more accurate for judging mobile robot poses. As the mobile robot position is updated, it will resample particles. Moreover, the particles with larger weights will remain. After many iterations, the results show that it will be closer to the actual mobile robot pose. This method achieves precise position and reduces the complexity of the mobile robot. In short, this algorithm uses a particle filter to track the mobile robot's pose based on a known map.

### 2.3. Deep Learning

There are some layers in a basic deep learning model, including an input layer, multiple hidden layers, and an output layer. Because of the multiple hidden layers, deep learning models are called "deep". A deep learning model is different from a traditional neural network. In a deep learning model, the number of hidden layers is set as 20, 50, 100, 500, or even more. However, a traditional neural network contains only two to five layers. An architecture of a basic deep learning model is shown in Figure 9. Forward propagation and back propagation are two important and main procedures in a deep learning model. They are respectively illustrated in the following paragraphs.

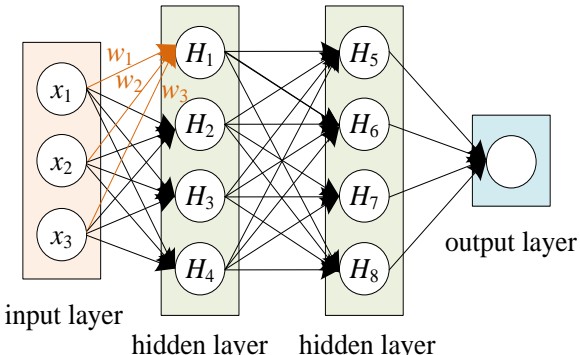

**Figure 9.** An architecture of a basic deep learning model.

For forward propagation, the initial weights are set up randomly. The values for the neurons in the first hidden layer can be calculated. In Figure 4, the value of $H_1$ with input node $x$, weight $w$, and a bias $b$ can be calculated through (17). The output of this equation is linear.

$$H_1 = x_1 * w_1 + x_2 * w_2 + x_3 * w_3 + b, \tag{17}$$

An activation function is applied to transfer these linear values into nonlinear. Before calculating the neurons in the second hidden layer ($H_5$, $H_6$, $H_7$, $H_8$), some activation functions are proposed, such as sigmoid, Rectified Linear Unit (ReLU), and so on. Then,

the forward propagation is processed until the output layer. The first set of predicted labels is generated.

The backward propagation starts after producing the first set of expected outcomes. Second, the loss and cost function between the expected labels and the annotated labels is determined. Several algorithms were suggested for loss and cost functions. The square loss function *l* is in (18), and it calculates the loss between each predicted label $f(x_i | \theta)$ and its annotated label $y_i$. The number of predictions is *n* and $\theta$ is the parameter of the model. The overall loss of predicted and annotated labels is further obtained by Mean Squared Error, as shown in (19).

$$l(f(x_i|\theta), y_i) = (f(x_i|\theta) - y_i)^2, \tag{18}$$

$$\text{MSE}(\theta) = \frac{1}{n} \sum_{i=1}^{n} (f(x_i|\theta) - y_i)^2, \tag{19}$$

### 2.3.1. CNN

In deep learning, CNN is a class of deep neural network, and it is the most common application in image and video recognition [22]. CNN contains one or more multiple convolutional layers, multiple pooling layers, and fully connected layer. Many image recognition models are also extended based on the CNN architecture. Figure 10 shows an example of a convolution operation on a two-dimensional tensor. The concept map of CNN is shown in Figure 11. The 2D convolution formula *S* in the convolutional neural network can be defined as (20), where the feature map is *I* and the kernel is *K*.

$$S(i,j) = (I * K)(i,j) = \sum_m \sum_n I(m,n)K(i-m,j-n), \tag{20}$$

where the indices *i* and *j* are concerned with the image matrices and those of *m* and *n* deal with that of the kernel.

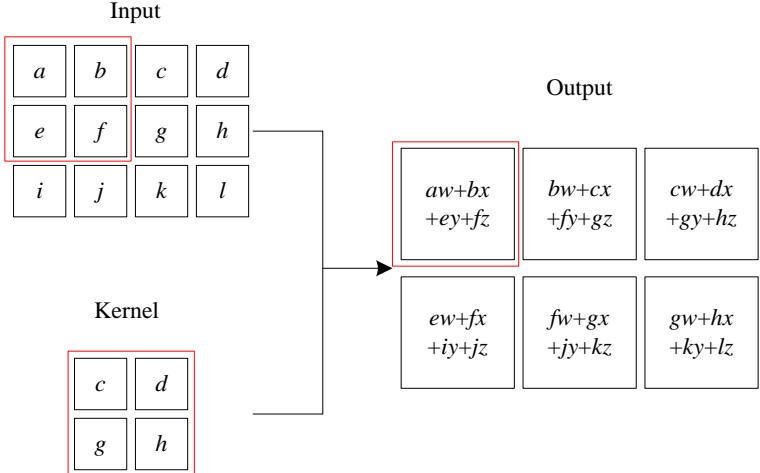

**Figure 10.** An example of two-dimensional convolution.

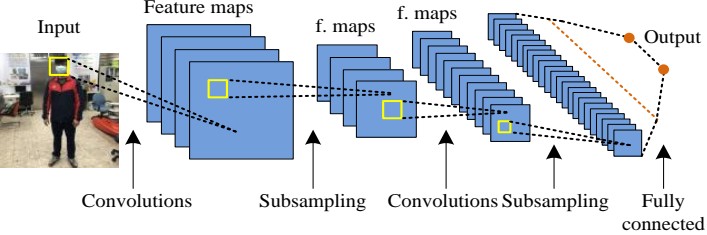

**Figure 11.** Concept map of CNN.

### 2.3.2. R-CNN

R-CNN is mainly applied for object detection. Object detection must not only achieve object classification, but also solve the problem of target positioning. That is, it must obtain the location information of the object in the original image. Therefore, it is necessary to explain the input data set of the bounding-box regression. The transformation formula is $f$, as in (21). The data set input to the bounding-box regression is $\{(P^i, G^i)\}_{i=1,\ldots,N}$, where the region proposal is $P^i = \left(P_x^i, P_y^i, P_w^i, P_h^i\right)$ and the ground truth is $G^i = (G_x{}^i, G_y{}^i, G_w{}^i, G_h{}^i)$. $\hat{G}_x, \hat{G}_y, \hat{G}_w, \hat{G}_h$ are the closer regression ground truth, as in (22) and (23). In Figure 12, the red box represents the candidate target box, the blue box is the target box predicted by the bounding box regression algorithm, and the green box is the real target box.

$$f\left(P_x^i, P_y^i, P_w^i, P_h^i\right) = (\hat{G}_x, \hat{G}_y, \hat{G}_w, \hat{G}_h) \approx (G_x, G_y, G_w, G_h),\tag{21}$$

$$\begin{cases} \hat{G}_x = P_w d_x(P) + P_x \\ \hat{G}_y = P_h d_y(P) + P_y \end{cases},\tag{22}$$

$$\begin{cases} \hat{G}_w = P_w \exp(d_x(P)) \\ \hat{G}_h = P_h \exp(d_y(P)) \end{cases},\tag{23}$$

where $x, y, w, h$ is the center point coordinates and width and height of bounding box.

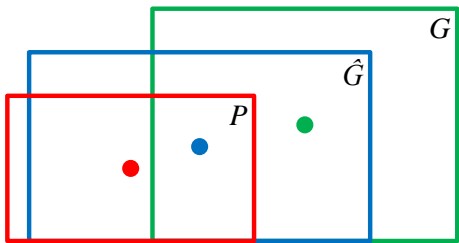

**Figure 12.** Bounding-box regression procedure.

The next thing to do is to solve $d_x$, $d_y$, $d_w$, and $d_h$. In the bounding-box regression, the linear regression is used in R-CNN, which represents features that were extracted from the fifth pooling layer in AlexNet and further to the fully connected layer. The predicted value $d_*(P)$ is in (24), where $*$ represents $x$, $y$, $w$, and $h$, and $\phi_5(P)$ is the eigenvector. The optimization of the loss function is $w_*$, as in (25), where $t_*$ is shown in (26). Hence, the least square method or gradient descent algorithm is used to obtain the result. The gradient descent function $x^{(t+1)}$ is shown in (27). In relative coordinates, the mapping relationship of bounding-box regression $f$ is as in (28). Equation (29) shows the reason why the bounding-box regression can be regarded as a linear transformation when the Intersection over Union (IoU) is large. When $x$ approaches zero, obtaining $\log(1+x) \approx x$, that is $\log(1+x)$ can be approximated as a linear transformation. Therefore, $t_w$ and $t_h$ of (26) can be rewritten as (30).

$$d_*(P) = w_*^T \phi_5(P),\tag{24}$$

$$w_* = \underset{\hat{w}_*}{\text{argmin}} \sum_{i=1}^{N} \left(t_*^i - \hat{w}_*^T \phi_5(P^i)\right)^2 + \lambda \|\hat{w}_*\|^2,\tag{25}$$

$$\begin{cases} t_x = \frac{G_x - P_x}{P_w} \\ t_y = \frac{G_y - P_y}{P_h} \\ t_w = \log \frac{G_w}{P_w} \\ t_h = \log \frac{G_h}{P_h} \end{cases},\tag{26}$$

$$x^{(t+1)} = x^{(t)} - \gamma \nabla f(x^{(t)}),\tag{27}$$

$$\begin{cases} f(\phi_1) = x_1 - p_1 \\ f(\phi_2) = x_2 - p_2 \end{cases}, \tag{28}$$

$$\lim_{x \to \infty} \frac{\log(1+x)}{x} = 1, \tag{29}$$

$$\begin{cases} t_w = \log \frac{G_w}{P_w} = \log \frac{G_w - P_w + P_w}{P_w} = \log(1 + \frac{G_w - P_w}{P_w}) \\ t_h = \log \frac{G_h}{P_h} = \log \frac{G_h - P_h + P_h}{P_h} = \log(1 + \frac{G_h - P_h}{P_h}) \end{cases}, \tag{30}$$

where $i$ is the currently calculated data and $N$ is the total amount of data. $\hat{w}_*$ is the parameter that needs to be learned. $t$ is the number of update parameters, and $\gamma$ is the learning rate. The features extracted by CNN are $\phi_1$ and $\phi_2$. $P_w$ and $P_h$ are the width and height of the region proposal. $G_w$ and $G_h$ are the width and height of the ground truth.

Using selective search, about 2000 region proposals can be found, and the extracted region proposals can be compressed to the same size. They are put into CNN to extract features, Support Vector Machine (SVM) is applied to segment and classify, and linear regression is carried out with bounding boxes. The flow is as follows:

1. A bunch of about 2000 possible regions is generated (region proposals).
2. Features are extracted through a pre-trained CNN model (e.g., AlexNet) and the results are stored.
3. The SVM classifier is used to distinguish whether it is an object or a background.
4. Finally, a linear regression model is used to correct the bounding-box position.

SVM is a supervised learning method. When using SVM, it must be with the kernel function so that SVM can get better performance in classification problems. A decision boundary is found in order to maximize the margins between the two categories so they can be perfectly separated. There are common kernel functions in (31) to (34).

Polynomial kernel function:

$$K(x_i, x_j) = (x_i \cdot x_j)^d, \tag{31}$$

$$K(x_i, x_j) = (1 + x_i \cdot x_j)^d, \tag{32}$$

Gaussian kernel function:

$$K(x_i, x_j) = \exp(-\frac{\|x_i - x_j\|^2}{2\sigma^2}), \tag{33}$$

Sigmoid kernel function:

$$K(x_i, x_j) = \tanh(k(x_i \cdot x_j) + \theta), \tag{34}$$

where $x_i$ and $x_j$ are vectors in the input space, $d$ is an integer, $\sigma$ is a free parameter, $\theta$ is coefficient, and $k$ is gamma.

First, the method of Selective Search is to draw the bounding box of the segment results. Next, a loop is used to merge the two boxes with the highest similarity each time until the entire image is merged into a single box. All the boxes in this process are the region proposals that are selectively searched. Figure 13 is the R-CNN architecture. Since R-CNN must generate about 2000 region proposals at first and each region must be put into CNN to extract features, there are many repetitive operations that take a lot of time.

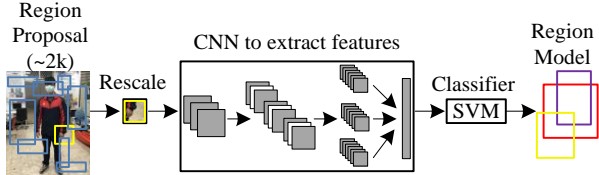

**Figure 13.** R-CNN architecture.

### 2.3.3. Mask R-CNN

Mask R-CNN is an extension of Faster R-CNN and was proposed by the same author. Faster R-CNN is a popular target detection framework, and Mask R-CNN extends it to an instance segmentation framework.

Faster R-CNN is mainly composed of two modules. The first layer is a deep full convolutional network to extract regions, and the second layer is a Fast R-CNN detector [23]. The Region Proposal Network (RPN) is the core of Faster R-CNN. The offset formula of the center coordinate $x$ and $y$ is as in (35). The loss function $L$ of Faster R-CNN is optimized for a multi-task, as shown in (36) and smooth$_{L1}$ is shown in (37).

$$\begin{cases} x = (t_x \times w_a) + x_a \\ y = (t_y \times h_a) + y_a \end{cases}, \tag{35}$$

$$\begin{aligned} L(\{p_i\}, \{t_i\}) &= \frac{1}{N_{\text{cls}}} L_{\text{cls}}(p_i, p_i^*) \\ &+ \lambda \frac{1}{N_{\text{box}}} \sum_i p_i^* \cdot \text{smooth}_{L_1}(t_i - t_i^*) \end{aligned} \tag{36}$$

$$\text{smooth}_{L_1} = \begin{cases} 0.5x^2, & \text{if}\,|x| < 1 \\ |x| - 0.5, & \text{otherwise} \end{cases}, \tag{37}$$

where $t$ is a vector and it is a bounding-box coordinate. $w$ and $h$ are width and height. $a$ is anchor. $p_i$ is the predicted probability of anchor $i$ being an object and $p_i^*$ is the ground truth label of whether anchor $i$ is an object. $t_i$ is predicted four parameterized coordinates and $t_i^*$ is the ground truth coordinates. $L_{\text{cls}}$ is the log loss function over two classes, $N_{\text{cls}}$ is set to be the mini-batch size, and $N_{\text{box}}$ is set to the number of anchor locations. $\lambda$ is a balancing parameter.

Mask R-CNN is a two-stage framework. The first stage is to scan the image and produce suggestions, and a target may be found in the regions. The second stage categorizes suggestions and creates bounding boxes and masks. Region of Interest Align (RoI Align) is proposed in Mask R-CNN. It removes the fixed-point process in the RoI pooling layer so that both the input proposal area and the coordinates of its segmented area use real numbers [24]. The training loss function $L$ of Mask R-CNN can be described as in (38), and $L_{\text{box}}$ and $L_{\text{mask}}$ are both effective for positive RoI. Mask R-CNN continues the RPN part of Faster R-CNN. The segmentation error $L_{\text{mask}}$ is defined as the average binary cross entropy loss function, as shown in (39).

$$L = L(\{p_i\}, \{t_i\}) + (L_{\text{cls}} + L_{\text{box}} + L_{\text{mask}}), \tag{38}$$

$$L_{\text{mask}} = -\sum_y y \log(\hat{y}) + (1 - y) \log(1 - \hat{y}), \tag{39}$$

where $L_{\text{box}}$ is the bounding box loss function. $y$ is the labeled value and $\hat{y}$ is the predicted value.

### 2.3.4. YOLO

YOLO is the first one-stage detector in deep learning. In the object detection, YOLO basically decomposes the graph into many grid cells, then predicts two bounding boxes and the category probability in each grid cell. Finally, the threshold and Non-Maximum Suppression method are used to obtain the answer. It is assumed that the size of the input image is $100 \times 100$, and $C$ objects are detected. YOLO will divide the image into $S \times S$ grids, and the size of the tensor output by YOLO is $S \times S \times (B \times 5 + C)$. If $S = 5$, the image will be equally divided into $5 \times 5$ grids, and each grid cell is $20 \times 20$. The confidence score is shown in (40) [25].

$$\text{Confidence score} = P(\text{object}) * IoU_{\text{pred}}^{\text{truth}}, \tag{40}$$

where the existing probability of class-specific objects in the box and the fitness between the predicted box and the object are both taken into consideration.

The confidence score is in (41); it shows both the probability of that class and how well the box fits the object. In the test phase, each bounding box will get a set of class-specific *i* confidence score, as shown in (42). In the entire YOLO architecture, except for the last layer with linear output, each layer will be equipped with leaky rectified linear activation (leaky ReLU). The activation function $\phi(x)$ is (43).

$$
\begin{aligned}
& p(\text{Class}_i|\text{object}) * IoU_{\text{pred}}^{\text{truth}} \\
&= \frac{p(\text{Class}_i|\text{object})}{P(\text{object})} * P(\text{object}) * IoU_{\text{pred}}^{\text{truth}}
\end{aligned}
\tag{41}
$$

$$
\text{Confidencescore} = P(\text{Class}_i) * IoU_{\text{pred}}^{\text{truth}},
\tag{42}
$$

$$
\text{leakyReLU} : \phi(x) = \begin{cases} x, & \text{if} x > 0 \\ 0.1x, & \text{otherwise} \end{cases},
\tag{43}
$$

where *x* is the input to a neuron.

YOLOv3 is currently the most widely used technology for object detection. First, the bounding box is framed in the image to select the suspected region proposal, then the feature value analysis is intercepted and the information is classified in the bounding box. YOLOv3 is an improved version of YOLO; the output frame rate is faster than other object detection methods (Single Shot MultiBox Detector, Retina-Net, Object Detection via Region-based Fully Convolutional Networks). It is different from the previous detection model, and it applies a single neural network to the full image. YOLOv3 continues the practice of YOLOv2 and the bounding-box predicted formula *b* is in (44). The confidence score of the predicted objects is shown in (45).

$$
\begin{cases}
b_x = \sigma(t_x) + c_x \\
b_y = \sigma(t_y) + c_y \\
b_w = p_w e^{t_w} \\
b_h = p_h e^{t_h}
\end{cases},
\tag{44}
$$

$$
\text{Confidence score} = P(\text{object}) * IoU(b, \text{object}),
\tag{45}
$$

where $(t_x, t_y, t_w, t_h)$ is the predicted output of the model (network learning target). The coordinate offset of the cell (in the unit of cell side length) is $(c_x, c_y)$, $(p_w, p_h)$ is the side length of the preset anchor box, and $(b_x, b_y, b_w, b_h)$ is the final prediction, which are the center coordinates, width and height of bounding box.

YOLOv3 chose to train on the global area of the picture. While speeding up, it can better distinguish between the target and the background. However, for large objects, the background may also be included as part of the target. It is particularly effective for small and medium objects; as for the large objects, it is not satisfactory. For the predicted pictures, YOLOv3 uses end-to-end detection. The most important content of YOLOv3 is a large and rich deep convolutional neural network model. There are in total 53 fully connected convolutional layers, so it is also called Darknet-53. There are three loss functions in YOLOv3: Loss function *l* of box, class, and object, respectively, as in (46) to (48). The total loss function is shown in (49); it is the sum of (46) to (48). The binary cross entropy formula BCE is in (50). The YOLOv3 structure diagram is illustrated in Figure 14.

$$
\begin{aligned}
l_{\text{box}} = \lambda_{\text{coord}} \sum_{i=0}^{S^2} \sum_{j=0}^{B} 1_{i,j}^{\text{obj}} (2 - w_i \times h_i) \\
\left[ (x_i - \hat{x}_i)^2 + (y_i - \hat{y}_i)^2 + (w_i - \hat{w}_i)^2 + \left( h_i - \hat{h}_i \right)^2 \right]
\end{aligned}
\tag{46}
$$

$$
l_{\text{cls}} = \lambda_{\text{class}} \sum_{i=0}^{S^2} \sum_{j=0}^{B} 1_{i,j}^{\text{obj}} \sum_{c \in \text{class}} p_i(c) \log(\hat{p}_i(c)),
\tag{47}
$$

$$
\begin{aligned}
l_{\text{obj}} = \lambda_{\text{nobj}} \sum_{i=0}^{S^2} \sum_{j=0}^{B} 1_{i,j}^{\text{nobj}} (c_i - \hat{c}_i)^2 \\
+ \lambda_{\text{obj}} \sum_{i=0}^{S^2} \sum_{j=0}^{B} 1_{i,j}^{\text{nobj}} (c_i - \hat{c}_i)^2
\end{aligned}
\tag{48}
$$

$$loss = l_{\text{box}} + l_{\text{cls}} + l_{\text{obj}}, \tag{49}$$

$$\text{BCE}(\hat{c}_i, c_i) = -\hat{c}_i \times \log(c_i) - (1 - \hat{c}_i) \times \log(1 - c_i), \tag{50}$$

where $S$ is the grid size and $B$ is the number of bounding boxes. If box $j$ and cell $i$ are matched together, $1_{i,j}^{\text{obj}} = 1$; otherwise, it is 0. If box $j$ and cell $i$ are not matched together, $1_{i,j}^{\text{nobj}} = 1$; otherwise, it is 0. $(x_i, y_i)$ is the location of the centroid of the anchor box and $(w_i, h_i)$ is the width and height of the anchor box. The classification loss is $p_i(c)$, all $\lambda$ are the weight parameters, $\hat{c}_i$ is the labeled value, and $c_i$ is the predicted value.

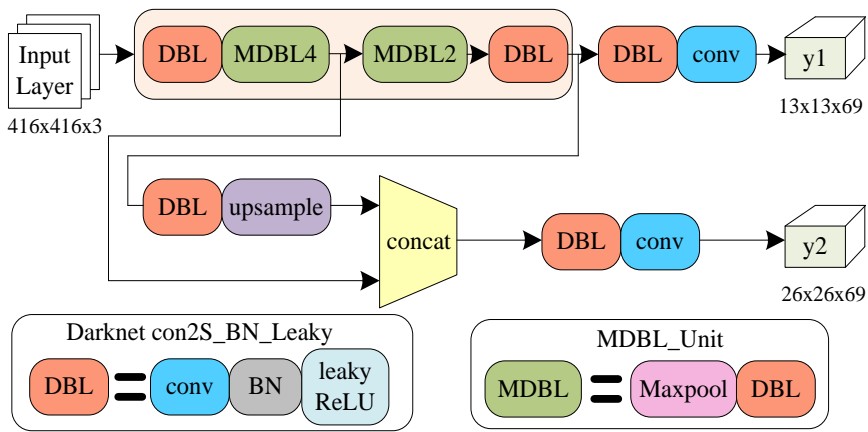

**Figure 14.** YOLOv3 structure diagram.

After considering various restrictions, YOLOv3 is more suitable for real-time detection on the mobile robot in this paper. When doing deep learning, concerning the memory size, this paper decides to apply a simple version; it is called Tiny-YOLOv3 [26]. Tiny-YOLOv3 also uses Darknet-53, but there are just seven convolutional layers [27]. While the detection accuracy of Tiny-YOLOv3 is not higher than YOLOv3, its speed raises dramatically. Therefore, this paper adopts YOLO to detect human faces.

## 3. Results

In this paper, the entire system architecture is composed of the following parts. First, the mobile robot is connected to the Raspberry Pi and the 3D LiDAR to construct a map through the SLAM algorithm in an unknown environment. Second, the *A** algorithm us used to plan the path, and fixed-point navigation is achieved via the AMCL algorithm in an indoor environment. Third, deep learning is applied for detection of temperature, mask wearing, and human faces during the movement of the mobile robot. Fourth, the detection results will be shown on the local side.

### 3.1. Hardware Design

In comparison to the system-on-a-programmable-chip (SoPC) methodology adopted for a pragmatic networked multirobot CPS in the field-programmable gate array chips [28], this paper proposes to use an autonomous mobile robot based on ROS, which is composed of some sensors, and its configuration is shown in Figure 15. This platform successfully combines software and hardware to achieve different task requirements, and it is verified in an indoor environment on the campus of National Taiwan University of Science and Technology.

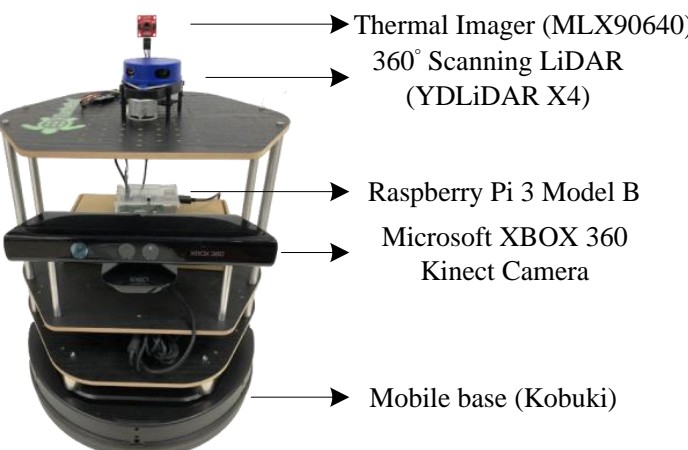

**Figure 15.** Autonomous mobile robot configuration.

Mobile Robot

This paper uses TurtleBot2 based on ROS as an experimental platform and an indoor mobile robot. TurtleBot2 is an advanced version of TurtleBot. The Turtle is derived from the Turtle robot, which in 1967 was powered by the Logo language of educational computer programming. TurtleBot is a low-cost, small-size, personal kit with open-source software and it is designed to let people who are new to ROS learn efficiently and teach computer programming language using Logo. In addition, the TurtleBot kit consists of a mobile base, 3D sensor, and the TurtleBot mounting hardware kit. It is also an ROS standard platform, and is popular among many developers and students.

The aim of TurtleBot2 is to reduce the size and expense of the platform without having to sacrifice its functionality and quality and it can also provide expandability at the same time. When TurtleBot2 is moving, it is a two-wheel differential drive composed of two independent driving wheels. The forward, backward, and turning movements of the mobile robot are done by adjusting the output power of its two motors. Moreover, it is possible for people to combine the mobile robot's hardware components (Kobuki base, Kinect camera) and embedded system with TurtleBot software installed to make applications. In this way, the entire mobile robot's architecture can be more complete.

### 3.2. Embedded System

The mobile robot needs to detect in a real indoor environment, so it must be capable of computed, analyzed, and applied data in real time. In order to save time and reduce transmission costs, computation and data storage are directly brought to the location needed without sending data to the cloud server.

In this paper, the deep learning model is trained by NVIDIA GEFORCE RTX Super 2060 and deep learning frameworks are adopted, such as TensorFlow. After obtaining the deep learning models, they are deployed to an embedded system, which is Raspberry Pi 3 Model B. There are a lot of compatible software on Raspberry Pi, so it uses it as the main control board.

### 3.3. Sensor

The distance measurement of an object by optical radar is identical to what is generally referred to as the radar, which measures the distance of an object by calculating the time interval between the transmitted and received pulse signals. LiDAR is one of the most popular sensors in the digital market today, so this paper applies it as a SLAM sensor. The LiDAR acronym means Light Detection and Ranging. The light emitted by LiDAR will be transmitted to the target, so it is a kind of remote sensing technology. It is unlike rangefinder that can only scan 240 degrees as in Figure 16. LiDAR can scan 360 degrees to

measure the distance from itself to the target because the speed of light is constant. Hence, the distance of the object can be known by this technology.

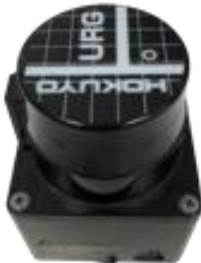

**Figure 16.** Laser rangefinder (Hokuyo URG-04LX-UG01).

This paper is to solve the indoor localization problem. Based on the advantages of lightness and portability, LiDAR has attracted more and more attention. It is also provided with a high angular resolution, high sampling rate, good range detection, and high robustness against environmental variability. After consideration and experiment confirmation, the result of LiDAR is better than general radar. Therefore, LiDAR is adopted as an indoor positioning sensor. Figure 17 is LiDAR (YDLiDAR X4).

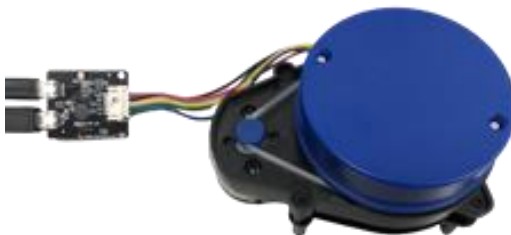

**Figure 17.** LiDAR (YDLiDAR X4).

For people's temperature detection, the thermal camera is used (MLX90640); it is shown in Figure 18. Inter-Integrated Circuit (I$^2$C) is the communication for data transmission. Its features are illustrated in Table 1.

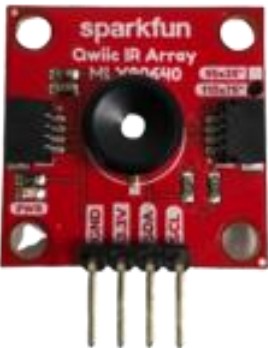

**Figure 18.** Thermal camera (MLX90640).

**Table 1.** Thermal camera (MLX90640) features.

| Specifications | Value |
|---|---|
| Operating Voltage | 3~3.6 V |
| Current Consumption | ~18 mA |
| Field of View | 110° × 75° |
| Measurement Range | −40~300 °C |
| Resolution | ±1.5 °C |
| Refresh Rate | 0.5~64 Hz |

*3.4. Experimental Results of SLAM Algorithms N*

3.4.1. Indoor Environment

This paper chose to create a map of the lab on the campus, which is shown in Figure 19, as the indoor experimental environment (library). While the space is not very large, it is enough for us to do every experiment and validation. During the experiment, the mobile robot was controlled to navigate a closing loop in the indoor environment and simultaneously scan around to generate an indoor map via a laser rangefinder and 3D LiDAR.

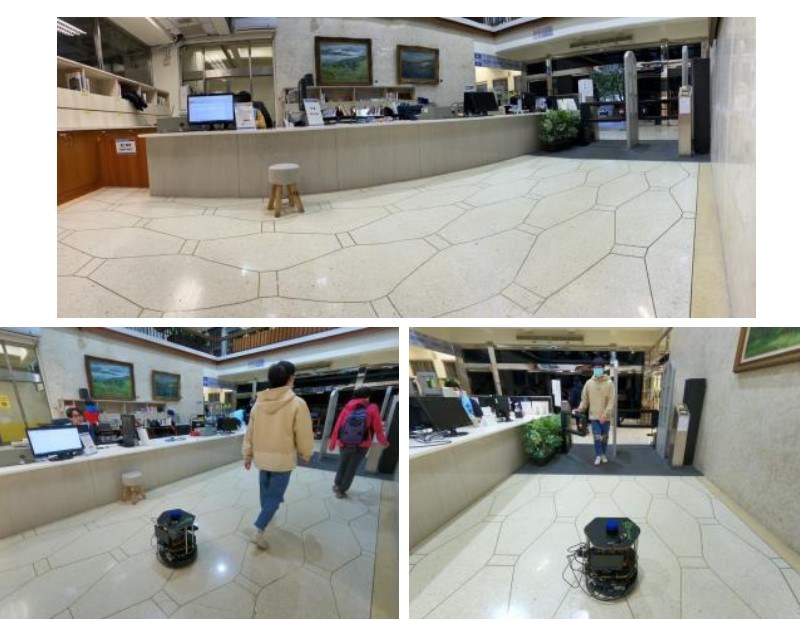

**Figure 19.** Indoor experimental environment (library).

3.4.2. Ground Truth-Based Indoor Map Is Built

The ground truth-based map was created by constructing the planes of walls and the floor from points obtained using LiDAR. The principle of LiDAR is that the pulsed laser light emitted by the laser transmitter is emitted from the robot to the ground and hits the buildings to cause scattering and reflection. A part of the light waves will be backscattered back to the receiver on the carrier, and the light signals will be converted into electrical signals and recorded. At the same time, the timer records the time from emission to reception of the same pulsed light signal. In this way, the distance from the robot to the target can be obtained.

3.4.3. Map from ROS-Based SLAM Algorithm and Map Comparison Is Built

This paper used three SLAM algorithms separately, Gmapping, Hector SLAM, Google Cartographer, in the same condition and the same indoor environment. When the mobile robot executed the SLAM algorithm, the map was saved. At the same time, ROS bag files recorded all the ROS topics from the mobile robot platform. Finally, these constructed maps were saved as the occupancy grid form and pgm format.

It is seen very clearly in Figures 20–25 that they are constructed by using laser rangefinder (Hokuyo URG-04LX-UG01) and 3D LiDAR (YDLiDAR X4) to perform three SLAM algorithms separately. The following created map results show that the effect of using LiDAR to build the indoor map is better and more complete than that of the laser rangefinder. The reason is that LiDAR scans are more comprehensive.

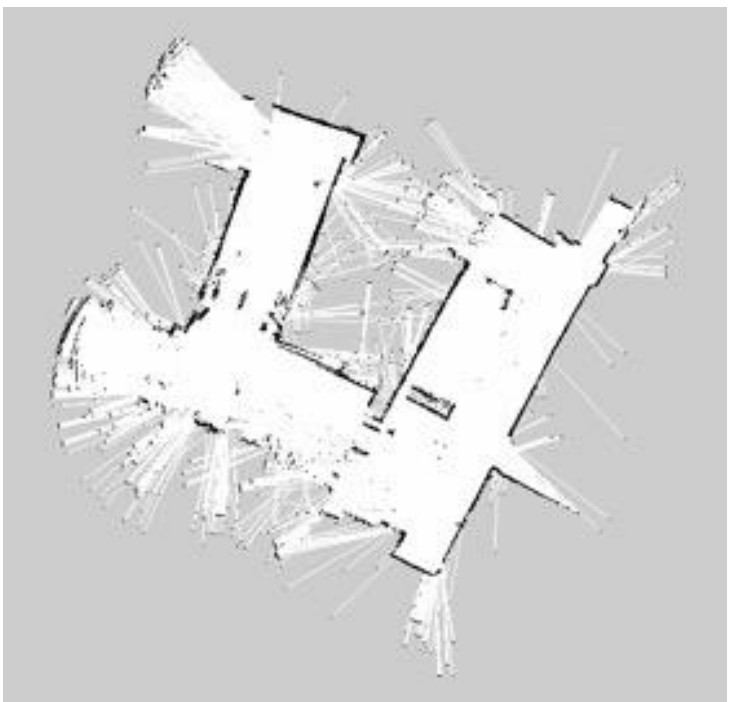

**Figure 20.** The map using Gmapping (Hokuyo laser rangefinder).

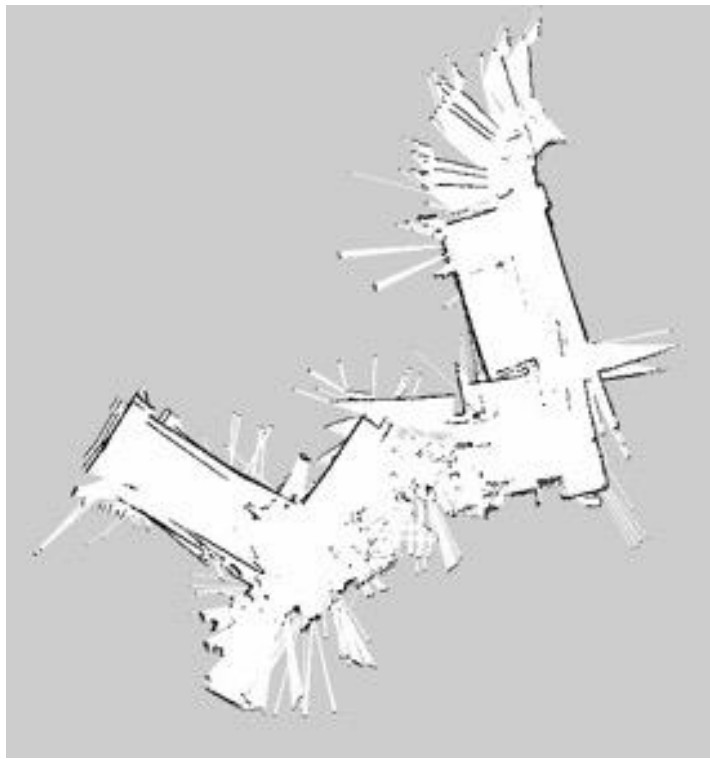

**Figure 21.** The map using Hector (Hokuyo laser rangefinder).

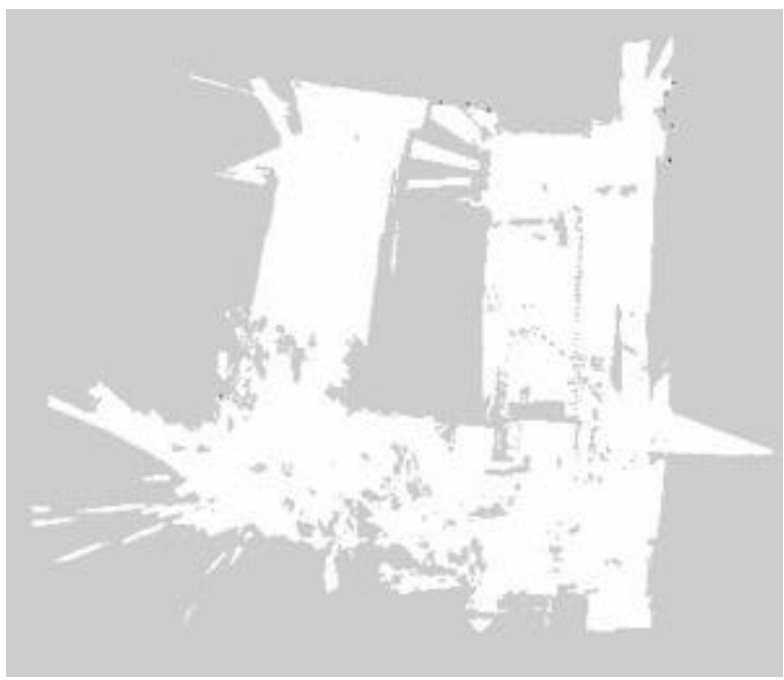

**Figure 22.** The map using Cartographer (Hokuyo laser rangefinder).

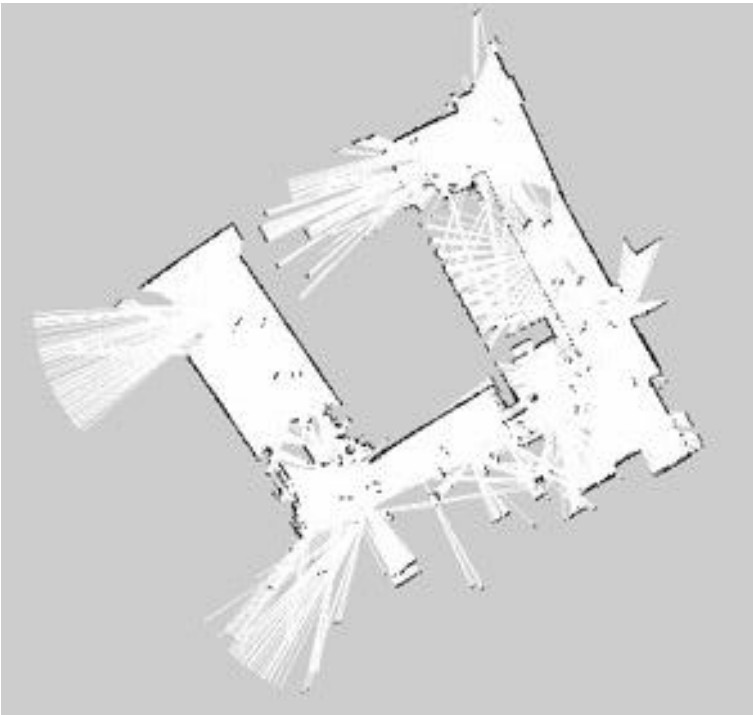

**Figure 23.** The map using Gmapping (YDLiDAR X4).

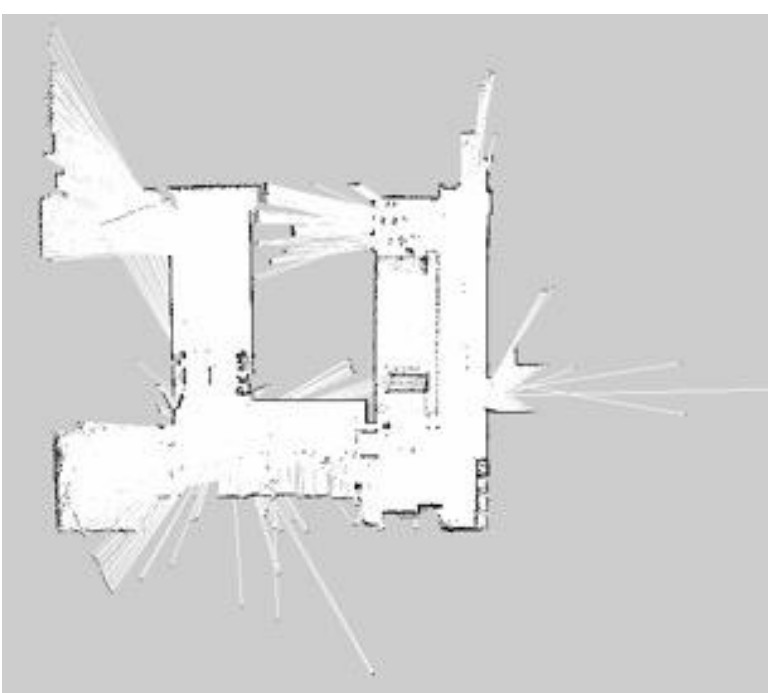

**Figure 24.** The map using Hector (YDLiDAR X4).

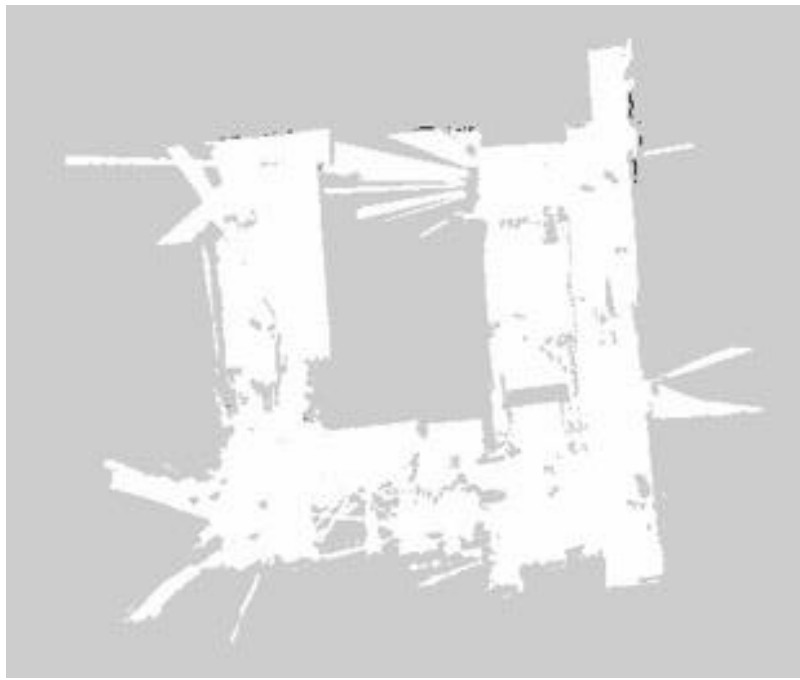

**Figure 25.** The map using Cartographer (YDLiDAR X4).

While the mobile robot was equipped with LiDAR to build indoor maps, bag files were recorded with the contents of their odometry. The data of these recorded bag files can be drawn as in Figures 26–28. The contents of bag files can also be summarized via the command-line "rosbag info".

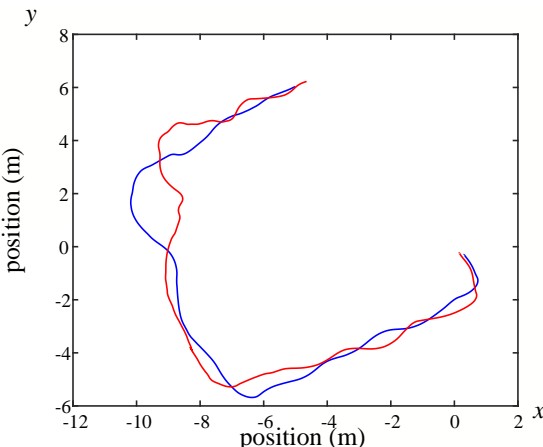

**Figure 26.** The odometry of using Gmapping (Red path is real path and blue path is predicted path).

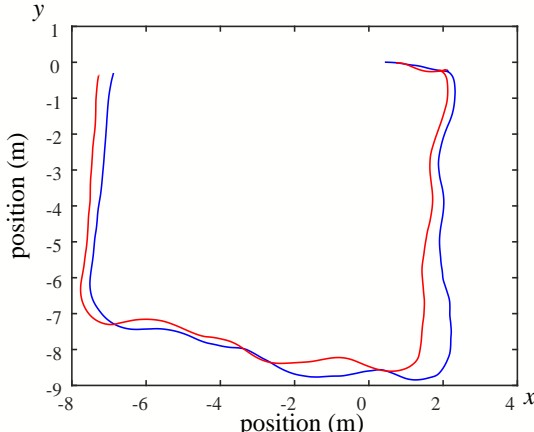

**Figure 27.** The odometry of using Hector (Red path is real path and blue path is predicted path).

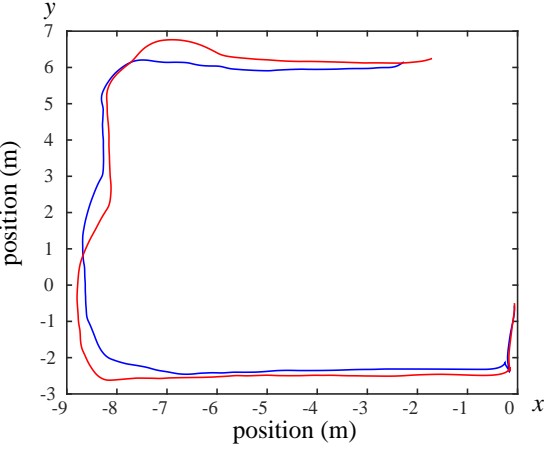

**Figure 28.** The odometry of using Cartographer (Red path is real path and blue path is predicted path).

*3.5. Experimental Results of Deep Learning*

3.5.1. Temperature Detection

For people's temperature detection, I$^2$C is used to do data transmission. I$^2$C is a synchronous, multi-master, multi-slave, packet switched, single-ended communication bus. It's typical voltage is +3.3 V and it uses two bidirectional open collector or open drain lines, which are Serial Data Line and Serial Clock Line. Because detecting people's temperature cannot use the ordinary camera, this paper uses the thermal camera, which

is MLX90640, to obtain people's temperature. I²C is the data bidirectional transmission. Detection screen displays the current heat map in real time, as shown in Figure 29. Through image pre-processing and using Mask R-CNN the face is framed and the temperature is displayed, as shown in Figure 30.

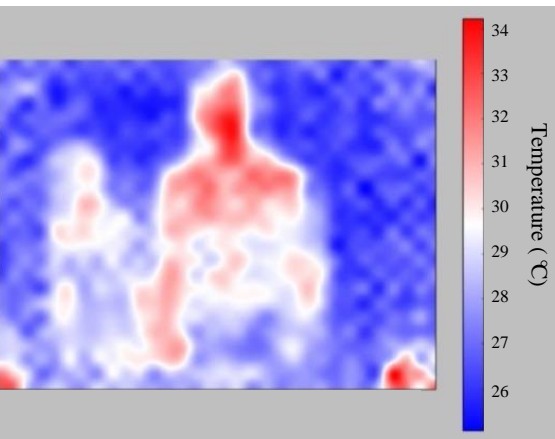

**Figure 29.** Current heat map in real time.

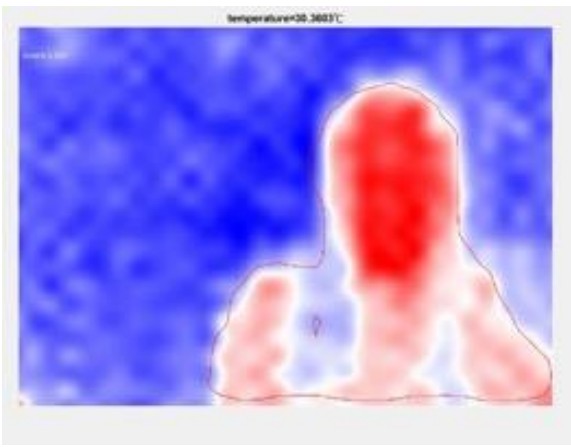

**Figure 30.** Display of people and temperature.

3.5.2. Mask Wearing Detection

In this paper, a mask wearing detector is trained with OpenCV, Keras, TensorFlow, and deep learning. For mask wearing detection, the whole process is divided into two phases. The first phase is to train the mask wearing detector and the second phase is to apply the mask wearing detector.

The dataset consists of about 1500 images, and they belong to two classes, which are with mask and without mask. The dataset of images was segmented into two sets, where 80% is the training set and 20% is the validation set. In order to implement the detection, the MobileNet V2 architecture is fine-tuned. There are three steps to set up fine-tuning, which are loading MobileNet with pre-trained ImageNet weights, constructing the head of the model that will be placed at the base model, and freezing the base layers of the network, whereas the head layer weights will be tuned. Fine-tuning is a strategy to save considerable time. The results are shown in the following; Figure 31 plots the loss and Figure 32 plots the accuracy. Table 2 is the performance of the model, including precision, recall, and $F_1$ score. Table 3 is the confusion matrix of the trained model. The Precision–Recall curve (P-R curve) and the Receiver Operating Characteristic curve (ROC curve) are shown in Figures 33 and 34. The area of ROC curve is called Area Under the ROC Curve (AUC), and its value is 0.975. A larger AUC value represents a better classification effect. Figure 35 is the

RMSE curve. The sensitivity and specificity are 0.9927 and 0.9783, respectively. Figure 36 is mask wearing detection in real time.

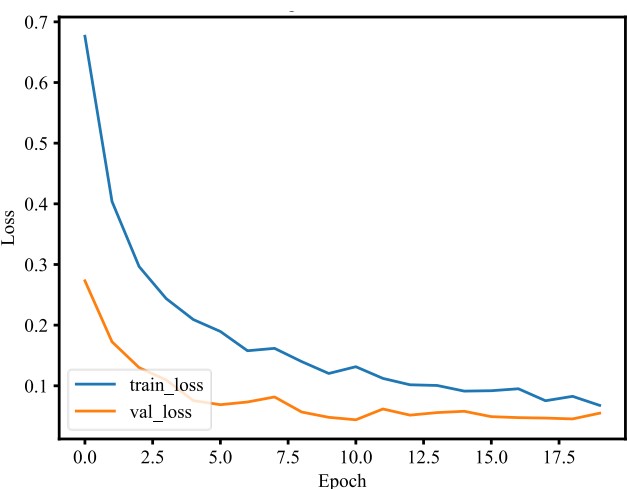

**Figure 31.** Loss of mask wearing detection.

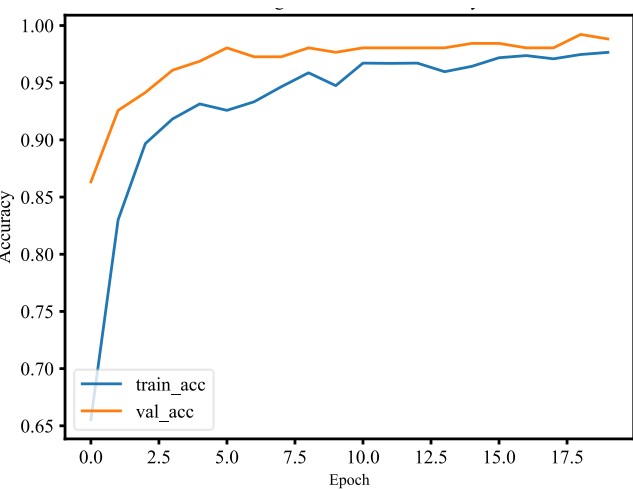

**Figure 32.** Accuracy of mask wearing detection.

**Table 2.** Performance of the model.

| Class | Precision | Recall | F1 Score |
|---|---|---|---|
| with mask | 0.98 | 1.00 | 0.99 |
| without mask | 1.00 | 0.98 | 0.99 |

**Table 3.** Confusion matrix of the model.

| | with Mask (Predicted) | without Mask (Predicted) |
|---|---|---|
| with mask (actual) | 136 | 1 |
| without mask (actual) | 3 | 135 |

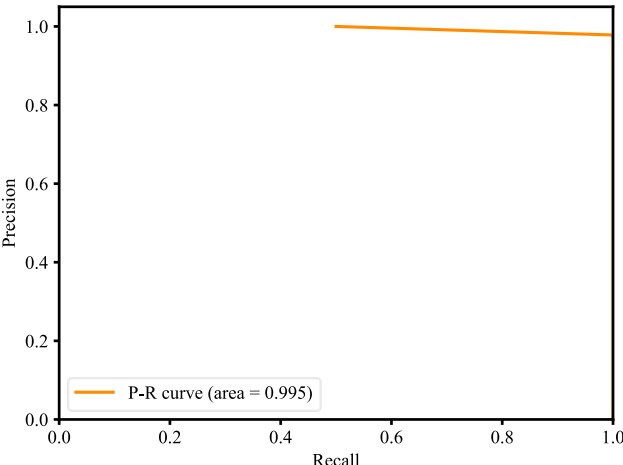

**Figure 33.** P-R curve of mask wearing detection.

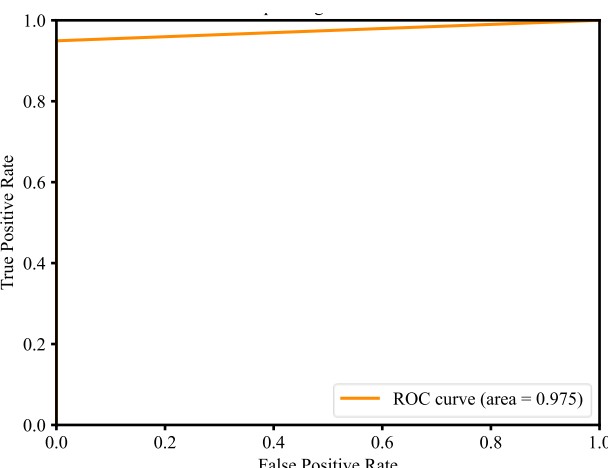

**Figure 34.** ROC curve of mask wearing detection.

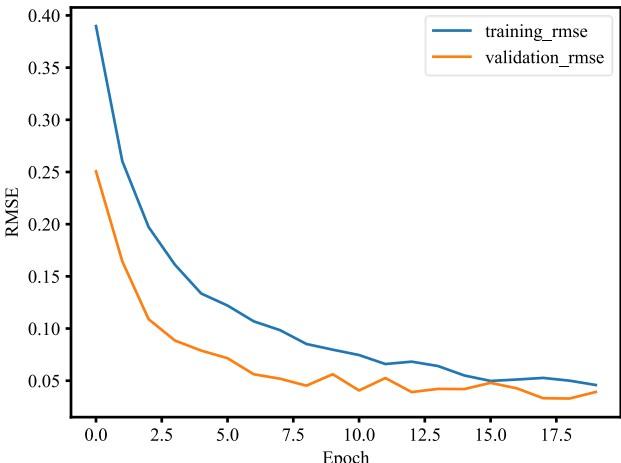

**Figure 35.** RMSE of mask wearing detection.

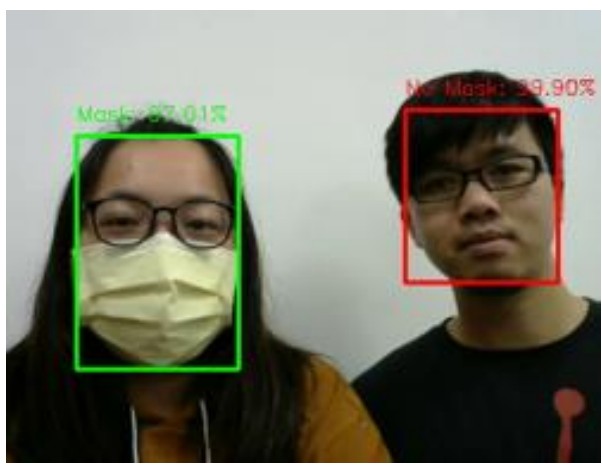

**Figure 36.** Mask wearing detection in real time.

### 3.5.3. Human Face Detection

The first method for human face detection is to apply OpenCV, Keras, TensorFlow, and deep learning to detect human faces, compute 128-dimension embeddings to quantify a face, train a MobileNetV2 model, and recognize faces in images and real time. There are three people and about 200 images for each person in the training data set. The training process includes the input data to the network and the triplet loss function. Each input batch of data has to include the anchor, positive image, and negative image. Scikit-learn's implementation of SVM is used to train the face detection model. After generating three different models, OpenCV is used to recognize human faces in real time and the frames per second value is 16.13. Figures 37 and 38 are training and validation loss and accuracy. Figure 39 is the P-R curve of multi-class. Figure 40 is the ROC curve of three models, and their AUCs are 0.83, 0.87, and 0.88, respectively. The RMSE curve is described in Figure 41.

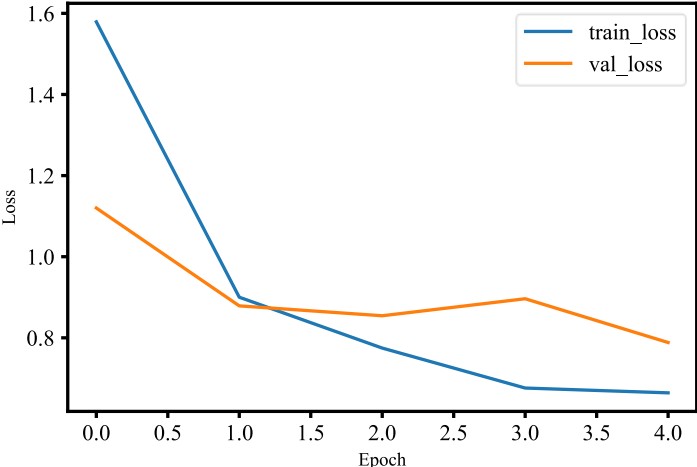

**Figure 37.** Loss of human face detection.

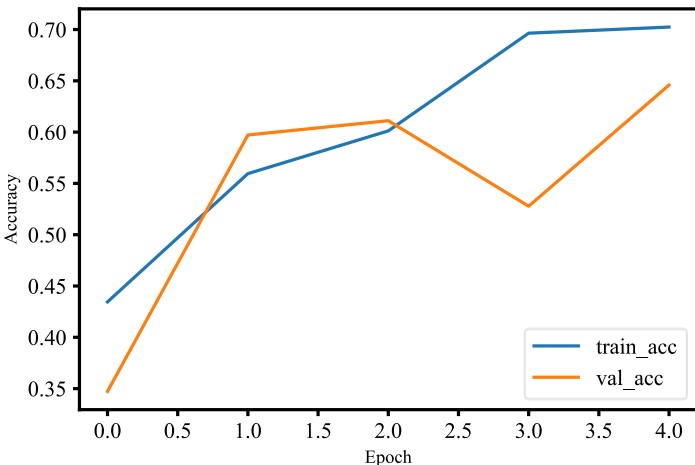

**Figure 38.** Accuracy of human face detection.

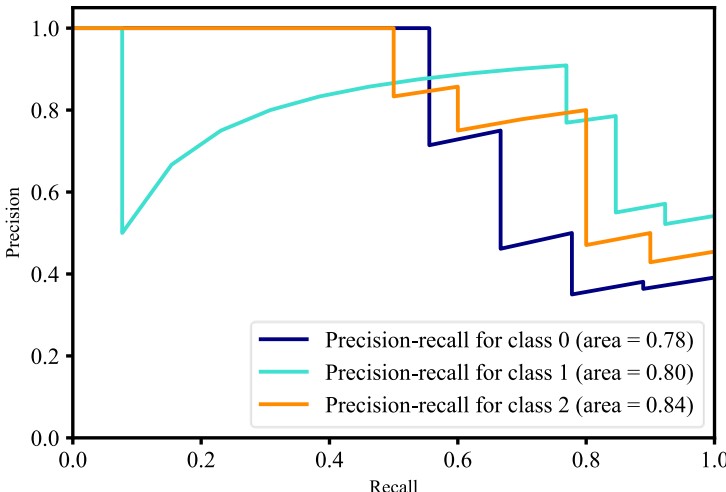

**Figure 39.** Multi-class P-R curve of human face detection.

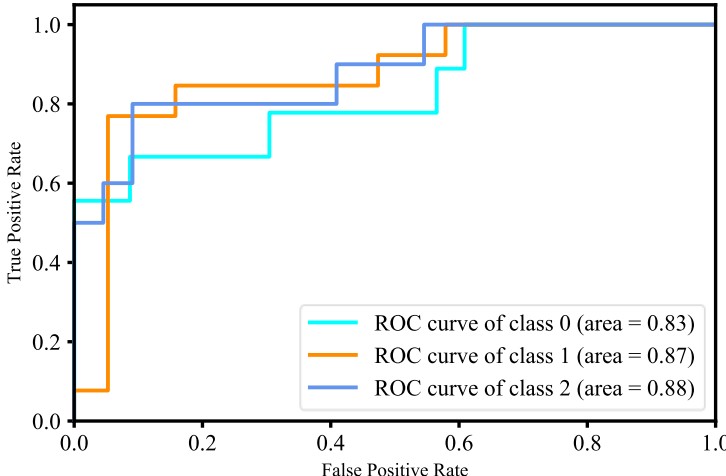

**Figure 40.** ROC curve of human face detection.

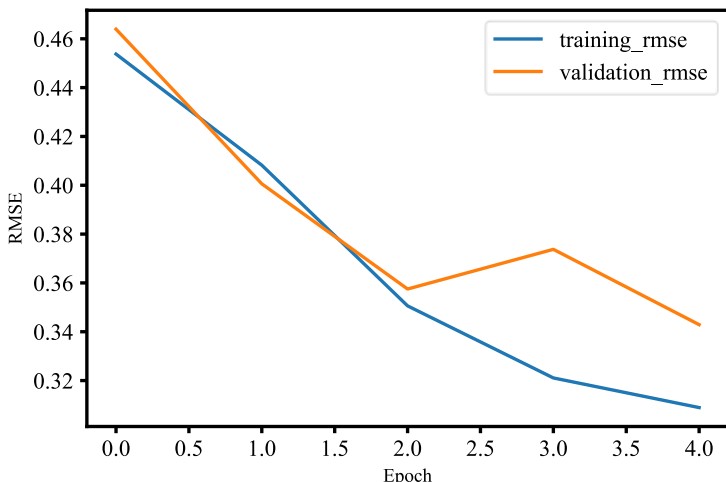

**Figure 41.** RMSE of human face detection.

The second method for human face detection uses the YOLO algorithm. It is built using dlib's state-of-the-art face detection built with deep learning. The difference from the above is the use of models trained by others. The model has a high accuracy, 99.38%. While the detection effect for the elderly and children is not very good, it is very little affected by light and different face angles. Hence, YOLO is the better algorithm for human face detection than the first method. Figure 35 is the human face detection in real time via YOLO.

## 4. Discussion

For building maps from ROS-based SLAM algorithms, three algorithms are used, namely Gmapping, Hector, and Google Cartographer. From Figures 20–25, it can be seen that the effect of using LiDAR to create a map is better than using a laser rangefinder. In addition, using LiDAR to build a map takes shorter time than using a laser rangefinder. Among these three SLAM algorithms, Gmapping and Google Cartographer have odometers. They use odometer data to correct the map and localize it. In Figures 20 and 23, the divergence around the map can be clearly seen. The algorithm with the best mapping effect is Google Cartographer, as shown in Figures 22 and 25. Figures 26–28 show the odometry of three different maps, and it can be seen that the closest actual path to the predicted path is the map built by Google Cartographer.

In the temperature detection part, a thermal camera is used to collect images for the dataset. First, images displayed by the thermal camera are accessed along with the corresponding temperature information. Second, all images are unified, which is called resizing. Third, the images are labeled, and the ground truth is generated. This step is to separate the person from the background. Then the image pre-processing is finished. Mask R-CNN is used to train the original images, temperature information, and labeled images. Three things are done at the same time during training, which are classification of people and background, human form detection, and framing of the human form. In the process of testing, after the human form is framed, the average temperature within the calculation range is the displayed temperature, as shown in Figure 30. Some errors in the displayed results will be affected by insufficient training data, imperfect label, and so on.

This paper used the pre-trained MobileNetV2 architecture with fine-tuning for mask wearing detection, where fine-tuning is the type of transfer learning that enables one to use pre-trained networks to recognize classes for which they were not initially trained. To fine-tune the pre-trained MobileNetV2 model, the following steps are used. First, we remove the fully connected layers (i.e., the head of the network) from the pre-trained model. Next, with random initializations, we replace the head with new fully connected layers, from which all layers below the head are frozen so that their weights cannot be

changed. Finally, we train the network so that the new fully connected layers will learn patterns from previously trained layers earlier in the network. During training process, we applied data augmentation to our datasets in order to enhance model generalization. The training and validation losses of the mask wearing detector model are less than 0.07, as shown in Figure 31. The validation loss is even smaller than the training loss, which indicates that our model generalizes very well. Similarly, the accuracy of training and validation of the mask wearing detector model are more than 97%, as shown in Figure 32. The validation accuracy is even better than the accuracy of the training, which also indicates that our model generalizes very well. For imbalanced datasets, the P-R curve is important to demonstrate that the model only concerns the correct prediction of the minority class, rather than correctly predicting the majority class. Precision is a ratio of the number of true positives divided by the sum of true positives and false positives, which defines how strong the positive class concerning being predicted by the model. Recall is the ratio of the number of true positives divided by the sum of the true positives and the false negatives, which determines how good the model is in predicting the positive class when the real outcome is positive. A skillful model is defined by a curve that bends in the direction of (1, 1), which means precision and recall equal to one. In Figure 33, our model also bends towards one, which represents that our model has good skills because of the balanced datasets. For a variety of different candidate threshold values between 0 and 1, the ROC curve is a plot of the false positive rate versus the true positive rate. The true positive rate is equal to the sensitivity that is implicitly equal to the recall. The ROC curve has valuable information about the problem of face detection and the skill of the model. The good model is represented at a point (0, 1). Our model is also at point (0, 1) and the AUC is 0.975, as shown in Figure 34. This implies that our model has good skills. RMSE is a measurement of the prediction error. A complete and good model is represented by having a very close RMSE from training and validation datasets. In Figure 35, we have approximately very close RMSE results, which indicates our model is good. In the end, OpenCV is used to achieve our trained mask wearing detection model in real time, which is shown in Figure 36. It can classify between human face with mask and without mask.

For face detection, similar to mask wearing detection, we used the pre-trained MobileNetV2 model with fine-tuning. Fine-tuning helps us to use our own training datasets to retrain the last layer of the pre-trained MobileNetV2 model. We added data augmentation to our datasets during training in order to improve generalization of our model. In Figure 37, the training and validation loss of the face detection model are less than 0.9 at epoch 2.0, and also the validation loss is smaller than the training loss for two epochs, which means that our model generalizes very well for two epochs. However, after two epochs of training, the model's validation loss begins to increase; that is, our model starts overfitting since our datasets are relatively not enough and similar to the original model datasets. We have used model check point to callback, which lets us monitor the training and validation dataset loss so that we can save the best weights. At epoch 2.0, the accuracy of the training and validation is more than 55%, which is shown in Figure 38. The accuracy of validation is better than training for the first two epochs, which means that for two epochs our model generalizes very well. The validation accuracy continues to decline after two epochs, which means our model begins overfitting. We used model check point to callback, which lets us monitor the training and validation datasets' accuracy to save the best weights. The P-R curve for three-class classification of our model is shown in Figure 39, and from the figure we know our model is able to accurately label class 2 as compared to other classes. The ROC curve for three-class classification of our model is shown in Figure 40, and the AUC of the ROC curve for class 0, class 1, and class 2 are 0.83, 0.87, and 0.88, respectively. In other words, our model has better skill concerning accurately labeling class 2 than other classes. The RMSE for three-class classification of our model is shown in Figure 41. In Figure 33, the training and validation RMSE values are very close up to two epochs, but the validation RMSE value is much higher than the training RMSE value after two epochs. This means our model begins to overfit because our datasets are relatively much less and similar to the

original datasets of the model. Finally, our trained face detection model was successfully implemented in real-time video streams with OpenCV. There is also an implementation for another form of pre-trained face detection, and it shows higher accuracy than our first model of face detection. This model of face detection is developed using dlib and deep leaning, which is YOLO. Dlib is a modern C++ toolkit containing algorithms for machine learning to solve real-world issues. The face detection model is able to accurately label a face in real time and in untrained images as shown in Figure 42.

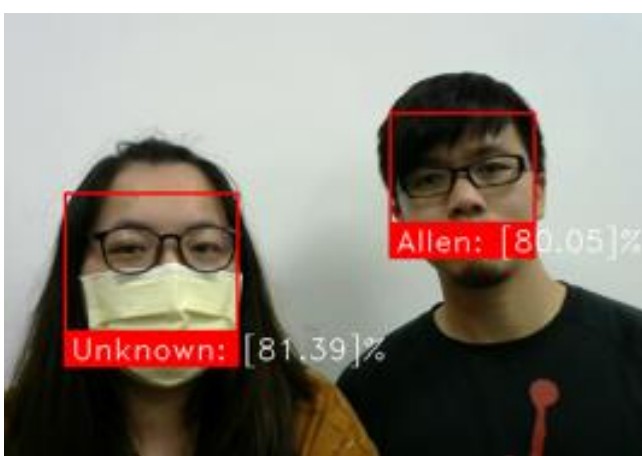

**Figure 42.** Human face detection in real time (YOLO).

## 5. Conclusions

After the various proofs of concept mentioned above, we summarize them separately. For building an indoor map, the constructed map via the Google Cartographer algorithm has the lowest error relative to the precise ground truth. At the same time, the mapping effect of LiDAR is better and more complete than the laser rangefinder. Therefore, these experiments proved that this SLAM algorithm has enough robustness for different mobile robot movements. In the process of building a map, the most critical point is how to correct the map and localize it. In these three SLAM algorithms, Gmapping and Google Cartographer have odometers. They use odometer data to correct the map and localize it. The Hector SLAM algorithm is different from the previous two algorithms. It only needs LiDAR data, not odometer data and does not provide the loop closure function. That is the reason why the result of Hector SLAM has lower accuracy. Hence, the result showed that building an indoor map using Google Cartographer with LiDAR is the best algorithm in this paper. For deep learning, there are three parts in total: Temperature detection, mask wearing detection, and human face detection. When doing deep learning, the frameworks used are Keras and TensorFlow. They used CNN, Mask R-CNN, and YOLO algorithms to achieve real-time detection.

In this paper, the wheeled mobile robot is differential drive. Different locomotion of robots can be made for different occasions in the future. More sensors can be installed on the mobile robot to achieve more functions. The extension of deep learning is to replace different algorithms. For human face detection, combining with the governments or school's database can confirm people's identities directly.

**Author Contributions:** Conceptualization, M.-F.R.L.; methodology, M.-F.R.L. and Y.-C.C.C.; software, Y.-C.C.C.; validation, M.-F.R.L. and Y.-C.C.C.; formal analysis, M.-F.R.L.; investigation, M.-F.R.L.; resources, M.-F.R.L.; data curation, Y.-C.C.C.; writing—original draft preparation, Y.-C.C.C.; writing—review and editing, M.-F.R.L.; visualization, Y.-C.C.C.; supervision, M.-F.R.L.; project administration, M.-F.R.L.; funding acquisition, M.-F.R.L. All authors have read and agreed to the published version of the manuscript.

**Funding:** This research was funded by [Ministry of Science and Technology (MOST) in Taiwan] grant number [108-2221-E-011-142-] and [Center for Cyber-physical System innovation from the Featured Areas Research Center Program within the framework of the Higher Education Sprout Project by the Ministry of Education (MOE) in Taiwan].

**Institutional Review Board Statement:** Not applicable.

**Informed Consent Statement:** Not applicable.

**Data Availability Statement:** Not applicable.

**Conflicts of Interest:** The authors declare no conflict of interest.

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
