# Peer review of "COVID-19 Pandemic Response Robot"

_machines, doi:10.3390/machines10050351_

Round 1

Reviewer 1 Report

Nice piece of work in general and written in an acceptable use of English language. The novelty of the paper is well derived in the text and the results are presented in an acceptable way. The paper is big and robust and contains all necessary information of the system implementation and its experimental operation characteristics. There are some modifications that you definitely need to do, though, as follows:

  • Avoid to use abbreviations in Abstract, even if you explain them later in the text. The way your typing stands now is mainly OK but for a reader who is not expert in the field, although highly interested to learn how robotics can held in the COVID-19 pandemic, will find it difficult for sure to follow the text. You should consider to make the abstract easier to read and understand.
  • In your introduction, you need to refer to 1 or 2 relevant research work from the close past that deal with mobile devices and/or platforms for totally different scopes. In this way, you will enrich your reference list and moreover you will support the introductory part and your research results more clearly. In addition, the motivation of your work will be better illustrated. For Example, you could try papers like “Action research implementation in developing an open source and low cost robotic platform for STEM education”, International Journal of Computer Applications, vol.178, issue 24, 2019 and Active Mapping and Robot Exploration: A Survey, Sensors 2021, 21, 2445.
  • In your reference list, refs 11 and 12 do not include the publication year, you should correct that.

Reviewer 2 Report

This paper integrated temperature detection, mask wearing detection, and human face detection. More sensors can be  installed on the mobile robot to achieve more functions. Reliable temperature measurement is not easy. It is a matter of sensors and measurement methodology. This applies to both the place and calculation of the measurement as well as the person history, e.g. what he did before the measurement, whether he was in the yard with a temperature of 40C or 0C. Moreover, temperature is not an objective indicator of CoVid disease. Mobility is also not very important, because you can make corridors for people who force the presence and correct body shape during the measurement. Assessing whether someone has a mask is trivial and does not require AI. However, I find the work interesting from a scientific point of view (comparison of several mapping methods and algorithms) and the work meets our expectations. It is very important to us that the diagnostic activities are performed by robots, especially if it poses a threat to the medical service team.
